# Spatial-temporal variations in riverine carbon strongly influenced by local hydrological events in an alpine headwater catchment

Xin Wang[1,2#], Ting Liu[1#], Liang Wang[1], Zongguang Liu[1,2], Erxiong Zhu[1,2], Simin Wang[1,2], Yue Cai[1,2], Shanshan Zhu[1,2], Xiaojuan Feng[1,2*]

[1] State Key Laboratory of Vegetation and Environmental Change, Institute of Botany, Chinese Academy of Sciences, Beijing, 100093, China

[2] College of Resources and Environment, University of Chinese Academy of Sciences, Beijing, 100049, China

[#] These authors contribute equally to this work.

*Correspondence to*: Xiaojuan Feng (xfeng@ibcas.ac.cn)

**Abstract.** Headwater streams drain > 70% of global land areas but are poorly monitored compared with large rivers. The small size and low water buffering capacity of headwater streams may result in a high sensitivity to local hydrological alterations and different carbon transport patterns from large rivers. Furthermore, alpine headwater streams on the "Asian Water Tower", i.e., Qinghai-Tibetan Plateau, are heavily affected by thawing of frozen soils in spring as well as monsoonal precipitation in summer, which may present contrasting spatial-temporal variations on carbon transport compared to tropical and temperate streams and strongly influence the export of carbon locked in seasonally frozen soils. To illustrate the unique hydro-biogeochemistry of riverine carbon in Qinghai-Tibetan headwater streams, here we carry out a benchmark investigation on the riverine carbon transport in an alpine small stream (the Shaliu River capturing headwaters within its catchment) based on annual flux monitoring, sampling at a high spatial resolution in two different seasons and hydrological event monitoring. We show that riverine carbon fluxes in the Shaliu River was dominated by dissolved inorganic carbon, peaking in the summer due to high discharge brought by the monsoon. Combining seasonal sampling along the river and monitoring of soil-river carbon transfer during spring thaw, we also show that both dissolved and particulate forms of riverine carbon increased downstream in the pre-monsoon season due to increasing contribution of organic matter derived from thawed soils along the river. By comparison, riverine carbon fluctuated in the summer, likely associated with sporadic inputs of organic matter supplied by local precipitation events during the monsoon season. Furthermore, using lignin phenol analysis for both riverine organic matter and soils in the basin, we show that the higher acid-to-aldehyde (Ad/Al) ratios of riverine lignin in the monsoon season reflect a larger contribution of topsoil likely via increased surface runoff compared with the pre-monsoon season when soil leachate lignin Ad/Al ratios were closer to those in the subsoil than topsoil solutions. Overall, these findings highlight the unique patterns and strong links of carbon transport in alpine headwater catchments with local hydrological events. Given the projected climate warming on the Qinghai-Tibetan Plateau, thawing of frozen soils and alterations of precipitation regimes may significantly influence the alpine headwater carbon transport, with critical effects on the biogeochemical cycles of the downstream rivers. The alpine headwater catchments may also be utilized as sentinels for climate-induced changes in the hydrological pathways and/or biogeochemistry of the small basin.

## 1. Introduction

Headwater streams, as the origins of stream network, comprise nearly 90% of the total length of global river networks (Downing et al., 2012) and drain > 70% of global land areas (Gomi et al., 2002). They are hence important sources of water, sediment, organic matter and nutrients for downstream regions (Gomi et al., 2002) and critical for maintaining ecological functions of the entire fluvial system including biogeochemical cycling (Biggs et al., 2017). However, headwater streams remain under-investigated in terms of biogeochemical processes and carbon transport patterns compared with large rivers so far, constraining an accurate understanding of the "boundless carbon cycle" (Battin et al., 2008).

Headwater streams differ from large rivers in several aspects that may affect their carbon transport. First, headwater streams have narrower width and shallower depth than large rivers, and are closely connected with adjacent terrestrial ecosystems via water and solute exchange (Battin et al., 2008; Öquist et al., 2014). Second, headwater streams have smaller drainage areas and buffering capacity for water flow than large rivers (Rivenbark and Jackson, 2004; Svec et al., 2005). In addition, water in headwater streams has a shorter residence time (Caillon and Schelker, 2020). The above characteristics collectively result in high sensitivity of headwater streams to local hydrological alterations or sporadic events such as precipitation (Benda et al., 2005; Richardson et al., 2005). Thus, riverine carbon transport in headwater streams may show a fast and strong response to local environmental variations and are receiving increasing attention (Gomi et al., 2002; Mann et al., 2015; Biggs et al., 2017) to improve our understanding of riverine carbon cycling (Flury and Ulseth, 2019; French et al., 2020; Battin et al., 2008).

Studies on headwater carbon transport have largely examined tropical and temperate streams in association with rainstorm events (Pereira et al., 2014; Argerich et al., 2016; Johnson et al., 2006). By comparison, headwater streams residing in alpine regions that are heavily affected by freeze-thaw and snow melting events are less well studied (Mann et al., 2015; Bröder et al., 2020; Chiasson-Poirier et al., 2020). The Qinghai-Tibetan Plateau, known as the "Asian Water Tower", harbors numerous headwater streams for seven major Asian rivers (Immerzeel et al., 2010; Qiu, 2008). This region is covered by large areas of glaciers, permafrost, and seasonally frozen soils (not underlain by permafrost layers), which are strongly affected by thawing with increasing temperatures in the pre-monsoon spring. Thawing of frozen soils and deepening of active layers can strongly affect catchment hydrology, including creating vertical and lateral flows, increasing soil filtration, enhancing groundwater-surface water exchange and baseflow (Song et al., 2019; Walvoord and Kurylyk, 2016). In addition, this region has a continental monsoon climate and is hence significantly affected by intense precipitation in the summer monsoon (Zou et al., 2017). Hydrological alterations induced by both spring thawing and summer monsoon likely result in unique seasonal patterns in riverine carbon transport in the Qinghai-Tibetan streams compared to headwater streams in other regions including the Arctic or tropical rivers (Zhang et al., 2013). The exported carbon could originate from both recently fixed modern carbon from terrestrial plants and aged carbon preserved in frozen soils within this region (Song et al., 2020; Qu et al., 2017), which show different decomposition characteristics (Mann et al., 2015) and thus may influence regional carbon cycling. However, the seasonal transport dynamics of these carbon pools along the Qinghai-Tibetan river continuum are relatively poorly

investigated. In particular, as one of the most climate-sensitive regions, warming-induced alterations of permafrost zone and precipitation patterns may change the hydrogeological characteristics of headwater streams (Chang et al., 2018) and thus affect riverine carbon transport. Yet, the variations of riverine carbon and its response to thawing and precipitation events in headwater streams on the Qinghai-Tibetan Plateau are poorly resolved.

70  To illustrate the unique hydro-biogeochemistry of riverine carbon in alpine headwater streams on the Qinghai-Tibetan Plateau, riverine carbon transports were investigated in a small stream (Shaliu River) feeding into the Qinghai Lake at three different levels. First, we provide a year-long biweekly monitoring record on the water discharge and concentrations of dissolved organic carbon (DOC) and dissolved inorganic carbon (DIC) to estimate monthly fluvial carbon fluxes for one year. Second, we use bulk concentration and biomarker (lignin phenols) analyses to examine riverine carbon composition and its

75 variations along the Shaliu River by sampling at a high spatial resolution in both pre-monsoon (spring thawing) and monsoon seasons. Third, to further identify the influence of hydrological events on the spatial-temporal variations of riverine carbon, we examine carbon transport from the adjacent soils to the stream during a 79-day thawing period and a short-term monsoon precipitation event. The characterization of lignin phenols, which are unique tracers of terrestrial plant derived organic matter (OM; Hedges and Mann, 1979) and provide useful information on the oxidation stage of terrestrial OM (indicated by lignin

80 phenol acid-to-aldehyde ratios; Bianchi and Canuel, 2011; Hedges et al., 1988), allows us to investigate the sources and transport of terrestrial OM in rivers. Based on these investigations, we hypothesize that the release of carbon from frozen soils during thawing events leads to a downstream increase of riverine carbon in the pre-monsoon season while carbon inputs through surface runoff in short-term precipitation events may result in a fluctuation of riverine carbon. Collectively, these investigations provide a benchmark illustration of riverine carbon transport in a headwater catchment on the Qinghai-Tibetan

85 Plateau.

## 2. Materials and Methods

### 2.1 Study area

  The Shaliu River, with a length of 110 km and a catchment area of 1442 $km^2$ (Zhang et al., 2013) and a mean annual daily discharge of 7.5 $m^3 s^{-1}$ during 2015 to 2016 (from Gangcha hydrological station), is the second largest stream flowing into the

90 Qinghai Lake on the northeastern edge of Qinghai-Tibetan Plateau at an altitude of 3200–3800 m above sea level (Figure 1a). The Shaliu River is a small stream (orders of 1−3) capturing the integrated signal across an alpine headwater catchment on the Qinghai-Tibetan Plateau. It flows through a semiarid alpine region widely covered by seasonally frozen soils (Wang et al., 2018) and is relatively well protected from human activities (Zhang et al., 2013; Cheng et al., 2018). The Shaliu River basin is covered by grassland (~71%) dominated by *Potentilla ansrina Rosaceae, Elymus nutans Griseb*, and *Deyeuxia arundinacea*

95 (Liu et al., 2018), bare land (16%) and wetland (10%; Cheng et al., 2018). The soils in the basin are mainly Gelic Cambisol (IUSS working group WRB, 2015) underlain by Triassic sandstone, late Cambrian metamorphic rocks (schist and gneiss), granites (Zhang et al., 2013) and a widespread distribution of late Paleozoic marine sedimentary rocks (Jin et al., 2009; Xiao

et al., 2013), and the latter rocks contain abundant limestone rich in carbonates (Bissell and Chilingar, 1967). The Shaliu River basin is under a continental monsoon climate characterized by warm, humid summers and cold, dry winters (Wang et al., 2018). The mean annual temperature is –0.5°C within the basin (Li et al., 2013) and mean annual precipitation is 370 mm, ~90% of which occurs in the monsoon season (June to September; Figure S1; Wu et al., 2016). The seasonally frozen soils in the watershed starts to thaw in late April to early May (i.e., the onset of pre-monsoon season) and refreezes in late October. The stream is partly or fully frozen from November to March.

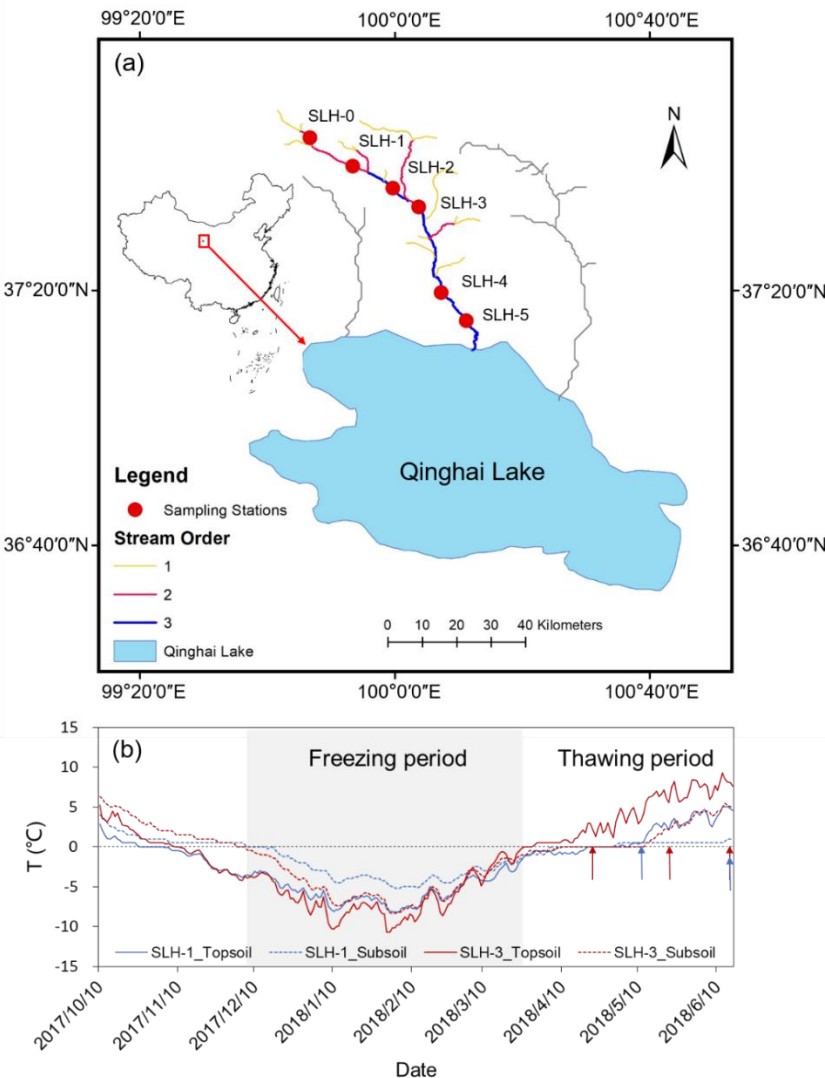

**Figure 1.** Sampling sites along the Shaliu River (a) and soil temperature (T) at the depth of 10 cm (referred to as topsoil) and 40 cm (subsoil) at SLH-1 and SLH-3 stations (b). The map in panel (a) is processed with ArcGIS 10.0. Stream order information is obtained from Lin et al. (2019). The blue and red arrows in panel (b) indicate the sampling time for soil solution at SLH-1 and SLH-3 during thawing event, respectively.

## 2.2 Sample collection

For annual flux assessment, the daily discharge of the Shaliu River was obtained from Gangcha hydrological station (SLH-4) where river water was collected biweekly from the middle of stream using pre-washed high-density polyethylene containers from May 2015 to April 2016. The high-density polyethylene containers were cleaned with soapy water, soaked in 10% HCl solution for 24 h, rinsed three times using Milli-Q water before drying, and re-rinsed three times with field river water prior to sampling. An aliquot of the water was immediately filtered through 0.45-μm acetate syringe filters and preserved

with 0.02% saturated mercury chloride ($HgCl_2$) in vials without headspace in the dark for DIC analysis. The remaining water was filtered through pre-combusted (550°C, 4 h) and pre-weighted 0.7-μm GF/F filters, preserved with a few drops of saturated $HgCl_2$, and kept frozen in acid-washed glass vials until DOC analysis.

      For seasonal water sampling, river water was collected using the same method as above at five evenly-distributed stations along the Shaliu River in May (pre-monsoon season) and August (monsoon season), 2015 (Figure 1a; SLH-5 was only sampled

in the pre-monsoon season). Water temperature, pH, dissolved oxygen (DO) and conductivity were measured in situ using a multi-parameter device (ProfiLine Multi 3320, WTW, Germany). Water was filtered and preserved as above for DIC and DOC analyses. Moreover, filtered water (0.7 μm) was kept frozen before measuring nitrogen (N) species and cations. The filters were freeze-dried for total suspended solids (TSS), particulate organic carbon (POC), particulate inorganic carbon (PIC), particulate N and particulate lignin phenol measurements. The remaining filtrates were acidified to pH 2 and extracted by solid

phase extraction (SPE) for dissolved lignin phenol measurements.

      To assess the influence of local hydrological events on riverine carbon, we further monitored spring thawing and a monsoonal precipitation. Specifically, soil temperature was monitored at 10 and 40 cm beneath the surface at SLH-1 and SLH-3 stations (~150 m away from the river bank) at 2-h intervals from October 2017 to June 2018 using iButton Temperature Loggers (DS1922L, Wdsen, China). Spring thawing occurred in late April when soil temperature was > 0 °C (Figure 1b). Soil

solutions were collected two or three times from top- (0–10 cm) and subsoil (30–40 cm) in April-June 2018 using pre-arranged ceramic head (0.2-μm), which is a porewater sampling device composed of a porous ceramic head connected with a tube for pulling vacuum and retrieving sample. Soil leachates were also directly collected by putting acid-washed carboys under soil layers in exposed soil profiles in June 2018 (Wang et al., 2018). Soil solutions and leachates were filtered (0.22-μm) immediately upon collection and preserved as previously described for DOC and dissolved lignin phenol analyses. For the

monsoonal precipitation, river water was sampled at 15-min intervals (four times) at SLH-4 station during a precipitation event (lasting for 1 hour) in August 2015 when a rainfall of 1.0 mm was recorded on the same day at the Gonghe weather station near Shaliu River basin, and filtered for the measurements of DOC, DIC, POC, PIC and TSS.

      To investigate connections between riverine and soil organic matter within the basin, topsoil (0–10 cm) and subsoil (40–60 cm) samples were collected from five locations 150–500 m away from the corresponding river sampling station in August

2015. At each location, three soil cores were collected from each of three random quadrats (1 × 1 m; intervals > 500 m) using a stainless steel gravity corer (diameter of 5 cm). Soils from the same depth and same quadrat were homogenized and shipped

back to laboratory. Freeze-dried soils were passed through a 2-mm sieve for soil organic carbon (SOC), soil N and soil lignin phenol measurements after removal of visible roots.

## 2.3 Bulk chemical analysis

The contents of riverine POC, particulate N, SOC and soil N were determined by elemental analyzer (Vario EL III, Germany) after fumigation with concentrated hydrochloric acid (HCl; Dai et al., 2019) and the analytical precision (standard deviation for repeated measurements of standards) was $\pm 0.1\%$. POC concentration (mg $L^{-1}$) was calculated based on POC content (%OC) and TSS concentration. PIC was calculated by subtracting POC from total particulate carbon quantified by elemental analyzer without fumigation.

The DOC, dissolved N and DIC concentrations were measured using a Multi N/C 3100 Analyzer fitted with an autosampler (Analytik Jena, Germany). Specifically, water samples for DOC analysis were manually acidified to pH < 2 with HCl, sparged automatically with oxygen ($O_2$) to remove inorganic carbon before DOC measurement via the high temperature catalytic oxidation procedure on Multi N/C 3100 Analyzer. For DIC, water samples were acidified on-line using phosphoric acid ($H_3PO_4$) with the $CO_2$ analysed by the non-scattering infrared detector on Multi N/C 3100 Analyzer. The analytical 155 precision was $\pm 1\%$ based on repeated measurements of standards. Inorganic N including ammonium ($NH_4^+$-N), nitrate ($NO_3^-$-N) and nitrite ($NO_2^-$-N) were determined colorimetrically on an Autoanalyser-3 (Bran & Luebbe, Germany). Dissolved organic N (DON) was calculated by subtracting all inorganic N from dissolved N. Major cations including calcium ($Ca^{2+}$) and magnesium ($Mg^{2+}$) were measured on an inductively coupled plasma mass spectrometer (ICP-MS, 7700X, Agilent, USA) after acidifying to pH < 2.

## 2.4 Lignin phenol analysis

Lignin phenols were analyzed using the alkaline copper oxide (CuO) oxidation method to trace terrestrially derived OM (Hedges and Ertel, 1982). Dissolved lignin phenols were concentrated with SPE cartridges containing 5 g of sorption materials composed of octadecyl carbon moieties ($C_{18}$) chemically bonded to a silica support ($C_{18}$-SPE Mega-Bond Elut; Agilent, USA). Cartridges were pretreated with methanol followed by acidified (pH 2) Milli-Q water. Filtered (through 0.7-μm pre-combusted 165 GF/F filter) river water was acidified to pH 2 using concentrated HCl, amended with pure methanol to a final concentration of 0.5% (v/v) to improve SPE efficiency (Spencer et al., 2010a), thoroughly mixed, and passed through the SPE cartridge with a peristaltic pump at a flow rate of 10 mL $min^{-1}$ (Louchouarn et al., 2010). The extraction efficiency of $C_{18}$ cartridge for lignin phenols was around 80-90% (Spencer et al., 2010a). All cartridges were dried and kept in a freezer in the dark until DOM was eluted with methanol and dried under nitrogen gas ($N_2$). Dried samples were mixed with 0.5 g CuO, 50 mg ammonium iron 170 (II) sulfate hexahydrate [$Fe(NH_4)_2(SO_4)_2 \cdot 6H_2O$] and 15 mg glucose (to avoid excessive oxidation), and soil solutions and leachates were mixed with the same reagents as well. All samples were mixed with 20 ml of 2 M $N_2$-purged sodium hydroxide (NaOH) solution in teflon-lined bombs at 170°C for 2.5 h. The lignin oxidation products were spiked with a recovery standard (ethyl vanillin), acidified to pH < 2 with 12-M HCl, and kept in the dark for 2 h. Oxidation products were extracted with ethyl

acetate three times, spiked with an internal standard (trans-cinnamic acid), and concentrated under $N_2$ for further analysis.

Lignin phenols, after converting to trimethylsilyl derivatives by reacting with N,O-bis-(trimethylsilyl)trifluoroacetamide (BSTFA) and pyridine, were quantified on a gas chromatograph-mass spectrometer (GC-MS; Thermo Fisher Scientific, USA) using a DB-5MS column (30 m × 0.25 mm inner diameter; film thickness 0.25 μm) for separation (Dai et al., 2019). Quantification was achieved by comparing with recovery standards (ethyl vanillin) to account for compound loss during extraction procedures. In addition, TSS and soil samples were extracted using the same procedures as mentioned above without

the addition of glucose.

Eight lignin-derived phenols, including vanillyl (V; vanillin, acetovanillone, vanillic acid), syringyl (S; syringaldehyde, acetosyringone, syringic acid), and cinnamyl (C; *p*-coumaric acid, ferulic acid) phenols, are used to represent the absolute ($\Sigma_8$; in the units of μg L$^{-1}$) and OC-normalized concentration ($\Lambda_8$; in the units of mg g$^{-1}$ OC) of lignin phenols. The acid-to-aldehyde (Ad/Al) ratios of V and S phenols are used to indicate lignin oxidation (Opsahl and Benner, 1995), which typically increase

with elevated degradation (Otto and Simpson, 2006) but may be affected by leaching and sorption processes as well (Hernes et al., 2007).

## 2.5 Carbon flux estimate from LOADEST

The daily discharge was combined with measured DOC and DIC concentrations at SLH-4 station to model their fluxes using the USGS load estimator (LOADEST) program (Runkel et al., 2004; https://water.usgs.gov/software/loadest/) within the

LoadRunner software package (URL: https://environment.yale.edu/loadrunner/). The best regression model with the lowest Akaike Information Criterion (AIC) and without variables for long-term change during the calibration period (i.e., Model 1, 2, 4, 6) within LOADEST was selected to fit the measured DOC (Equation 1) and DIC fluxes (Equation 2) for annual carbon flux estimate (Song et al., 2020; Tank et al., 2012; Zolkos et al., 2018; Zolkos et al., 2020):

$$\ln (\text{DOC flux}) = a_0 + a_1 \ln Q \tag{1}$$

$$\ln (\text{DIC flux}) = a_0 + a_1 \ln Q + a_2 \ln Q^2 + a_3 \sin(2\pi \text{dtime}) + a_4 \cos(2\pi \text{dtime}) \tag{2}$$

where flux is provided in kg day$^{-1}$, $Q$ is river discharge in cubic feet sec$^{-1}$ transformed from metric units, $\ln Q$ equals ln(streamflow) minus center of ln(streamflow), dtime equals decimal time minus center of decimal time, other parameters (i.e., $a_0$, $a_1$, $a_2$, $a_3$, $a_4$) are shown in Table S1. LOADEST can provide daily, monthly and annually fluxes using adjusted maximum likelihood estimation (AMLE) approach (Song et al., 2020; Tank et al., 2012).

## 200 2.6 Statistical analyses

Normal distribution of data and homogeneity of variance were checked for all variables using Shapiro-Wilk and Levene tests, respectively. Differences in river water parameters between the pre-monsoon and monsoon seasons were tested using independent-sample t-test. Differences in the Ad/Al ratios between top- and subsoils were analyzed by paired t-test. Differences in the Ad/Al ratios between soil solutions and leachates at SLH-1 station during thawing events were determined

using one-way ANOVA followed by post-hoc Duncan test. Differences in river water pH were checked by Mann-Whitney U

test due to non-normal distribution of data. Pearson correlation was used to assess relationships between DIC and cation concentrations and between riverine carbon concentration and the distance of sampling sites from SLH-0, where autocorrelation of riverine carbon concentrations among samples was not significant based on Durbin-Watson test. Differences and correlations were considered to be significant at a level of $p < 0.05$. All statistical analyses were conducted using SPSS 25 (SPSS, Chicago, USA).

## 3. Results

### 3.1 Discharges and carbon fluxes

The total annual discharge of Shaliu River at SLH-4 station was $2.4 \times 10^8$ m$^3$ during 2015 to 2016. Its daily discharge ranged from 0.25 to 82.40 m$^3$ s$^{-1}$, following a normal distribution with a maximum of 82.40 m$^3$ s$^{-1}$ in the monsoon season (Figure 2a). DIC concentration at SLH-4 station showed a large intra-annual variability from 20.8 to 50.9 mg L$^{-1}$ with a minimum value in the monsoon season, and had a negative relationship with river discharge (Figure 2a). On the contrary, DOC concentrations were relatively constant and fluctuated around 2.5−3.1 mg L$^{-1}$ (Figure 2a). Accordingly, the DIC and DOC fluxes at SLH-4 station were estimated to be 8.0 and 0.6 Gg year$^{-1}$ (1 Gg = $10^9$ g), respectively, both of which were highest in summer, accounting for 58-60% of the corresponding annual fluxes (Figure 2b). Moreover, annual DIC flux was more than 10 times larger than the DOC flux in the Shaliu River and constituted 90% of the total dissolved carbon flux.

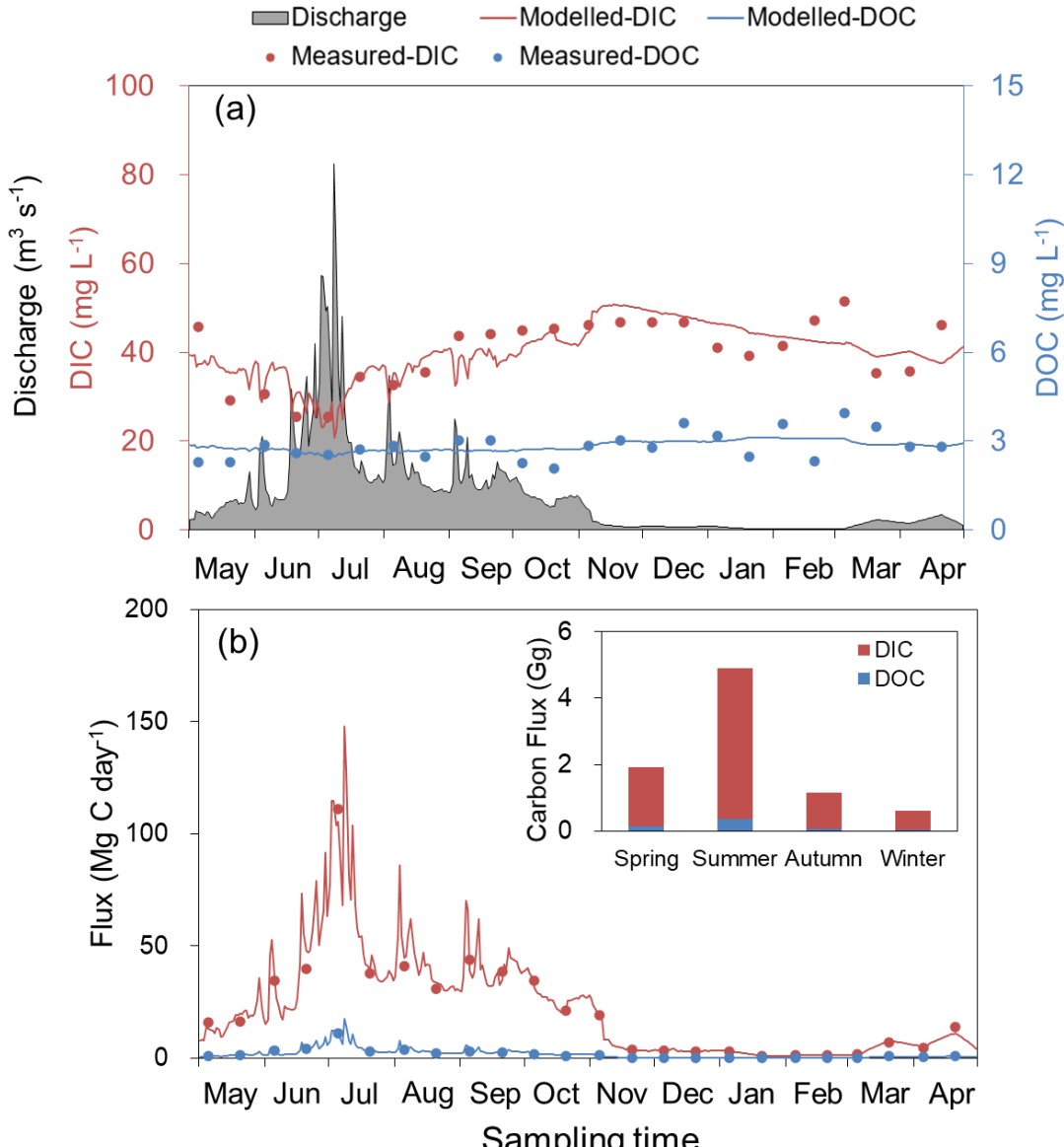

**Figure 2.** Discharge, dissolved inorganic carbon (DIC) and dissolved organic carbon (DOC) concentrations (a) and fluxes (b) exported from Shaliu River at SLH-4 station from May 2015 to April 2016. The modelled concentrations in (a) and modelled fluxes in (b) are derived from load estimator (LOADEST). The inserted columns in panel (b) show the seasonal variations of carbon fluxes classified as follows: spring (May to June), summer (July to September), autumn (October to November), and winter (December to the next April).

**Table 1.** River water properties along the Shaliu River.

| Sample | Lat. | Long. | E | T | pH | Cond. | DO | TSS | DIN | DON | Carbon content (POC) | DOC/POC | DOC/DON |
|---|---|---|---|---|---|---|---|---|---|---|---|---|---|
| | (°N) | (°E) | (m a.s.l.) | (°C) | | (µs cm$^{-1}$) | | | (mg L$^{-1}$) | | (%) | | |
| Pre-monsoon season (May, 2015) | | | | | | | | | | | | | |
| SLH-0 | 37.73 | 99.78 | 3846 | 10.4 | 8.5 | 369 | 6.2 | 13.2 | NA | NA. | 7.80 | 1.25 | NA |
| SLH-1 | 37.66 | 99.89 | 3678 | 8.2 | 8.4 | 267 | 6.4 | 15.2 | NA. | NA. | 8.29 | 1.09 | NA |
| SLH-2 | 37.60 | 100.00 | 3553 | 8.8 | 8.5 | 302 | 6.1 | 18.6 | NA | NA | 5.54 | 1.74 | NA |
| SLH-3 | 37.55 | 100.06 | 3485 | 7.7 | 8.4 | 417 | 6.8 | 23.4 | NA. | NA. | 7.48 | 1.14 | NA |
| SLH-4 | 37.33 | 100.12 | 3293 | 11.5 | 8.4 | 355 | 6.6 | 25.8 | NA | NA | 5.47 | 1.63 | NA |
| SLH-5 | 37.25 | 100.19 | 3243 | 12.9 | 8.2 | 342 | 5.9 | 33.4 | NA. | NA | 6.98 | 0.99 | NA |
| **Mean** | | | | **9.9a** | **8.4a** | **342a** | **6.3a** | **21.6a** | **NA** | **NA** | **6.92a** | **1.31b** | **NA** |
| Monsoon season (August, 2015) | | | | | | | | | | | | | |
| SLH-0 | 37.73 | 99.78 | 3846 | 8.8 | 7.6 | 339 | 5.3 | 5.2 | 0.8 | 0.3 | 5.00 | 9.39 | 10.9 |
| SLH-1 | 37.66 | 99.89 | 3678 | 10.0 | 7.7 | 305 | 5.3 | 6.2 | 1.2 | 0.4 | 7.26 | 4.89 | 6.8 |
| SLH-2 | 37.60 | 100.00 | 3553 | 9.3 | 7.5 | 313 | 5.8 | 22.4 | 1.5 | 0.4 | 4.96 | 2.43 | 7.3 |
| SLH-3 | 37.55 | 100.06 | 3485 | 8.0 | 7.5 | 377 | 5.9 | 9.8 | 3.0 | 0.5 | 5.20 | 4.32 | 5.2 |
| SLH-4 | 37.33 | 100.12 | 3293 | 17.9 | 8.4 | 355 | 4.4 | 26.8 | 1.6 | 0.3 | 5.63 | 1.63 | 10.5 |
| **Mean** | | | | **10.8a** | **7.7b** | **338a** | **5.3b** | **14.1a** | **1.6** | **0.4** | **5.61b** | **4.54a** | **8.2** |

Different lowercase letters indicate significant differences between the pre-monsoon and monsoon seasons ($p < 0.05$). Lat., Latitude; Long., Longitude; E, elevation; T, water temperature; Cond., conductivity; DO, dissolved oxygen; TSS, total suspended solid; DOC, dissolved organic carbon; POC, particulate organic carbon; DIN, dissolved inorganic nitrogen; DON, dissolved organic nitrogen; DOC/POC, ratio of DOC to POC; DOC/DON, the atomic ratio of DOC to DON; a.s.l., above sea level; NA, not analyzed.

### 3.2 River water properties and lignin phenols

River water was slightly alkaline with a higher pH values in the pre-monsoon ($8.4 \pm 0.04$) than monsoon season ($7.7 \pm 0.2$) in the Shaliu River ($p < 0.05$; Table 1). DO concentration was also higher in the pre-monsoon (5.91–6.76 mg L$^{-1}$) than monsoon season (4.39–5.87 mg L$^{-1}$; $p < 0.05$). Consistent with the annual monitoring results, DIC was the dominant carbon species, accounting for approximately 90% of the total carbon, followed by DOC, POC and PIC, respectively (Figure S2a). DOC concentration was higher in the monsoon than pre-monsoon season, while both PIC and POC concentrations were higher in pre-monsoon season ($p < 0.05$; Figure S2a) due to the higher TSS in majority sampling sites (Table 1). The DOC/POC ratio was lower in the pre-monsoon than monsoon season ($p < 0.05$; Table 1). The POC concentration was positively related to TSS ($p < 0.05$; Table 1) and DIC concentration had positive correlation with cation (Ca$^{2+}$ + Mg$^{2+}$) in both seasons ($p < 0.05$; Figure S2b). Both dissolved (i.e., DOC and DIC) and particulate carbon (i.e., POC and PIC) concentrations fluctuated along the Shaliu River in the monsoon season, while most of them increased significantly downstream in the pre-monsoon season ($p < 0.05$; Figure 3a) except for a marginally significant increasing trend for POC ($p = 0.07$; Figure S3).

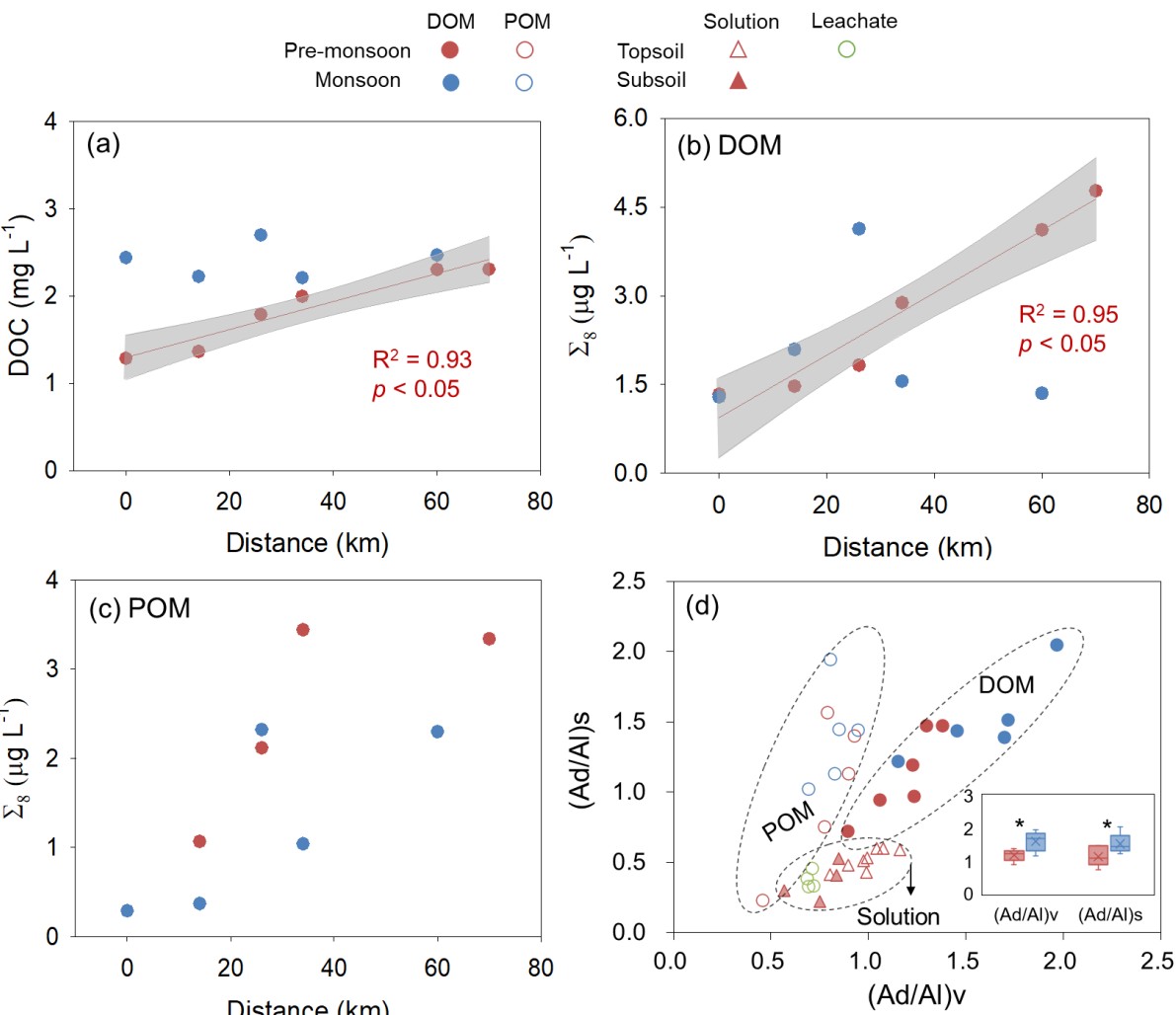

**Figure 3.** Variations of dissolved organic carbon (DOC) in Shaliu River water (a), absolute concentration of lignin phenols ($\Sigma_8$) in riverine dissolved organic matter (DOM; b) and particulate organic matter (POM; c) during the pre-monsoon and monsoon seasons in 2015, the acid-to-aldehyde (Ad/Al) ratios of syringyl (S) and vanillyl (V) phenols in the riverine DOM, POM, soil solutions and leachates (d). The abscissa in panel (a), (b) and (c) mean the distance of sampling sites from SLH-0. The red lines in panel (a) and (b) correspond to the linear regression of data ($p < 0.05$), and the grey shaded regions in panel (a) and (b) show 95% confidence intervals. The inserted box in (d) is the comparison of $(Ad/Al)_V$ and $(Ad/Al)_S$ ratios of dissolved lignin phenols between pre-monsoon and monsoon seasons, respectively, with asterisks indicating significant differences (independent sample t tests, n = 5, $p < 0.05$). The solid bar and cross in the inserted box mark the median and mean of each data set, respectively. The upper and lower ends of box denote the 0.25 and 0.75 percentiles, respectively.

The absolute concentration of dissolved lignin phenols ($\Sigma_8$) increased longitudinally from 1.33 µg L$^{-1}$ at SLH-0 to 4.77 µg L$^{-1}$ at SLH-5 in the pre-monsoon season, while it peaked at SLH-2 (4.13 µg L$^{-1}$) and decreased to 1.35 µg L$^{-1}$ in the

monsoon season (Figure 3b). The absolute concentration of particulate lignin phenols showed similar trends with dissolved lignin, which increased downstream from 1.07 to 3.44 µg L$^{-1}$ in the pre-monsoon season and peaked at SLH-2 in the monsoon

season. The (Ad/Al)$_S$ ratios showed no significant difference between dissolved and particulate lignin phenols, while the (Ad/Al)$_V$ ratios of dissolved lignin phenols ranging from 0.89 to 1.96 were significantly higher than that in particulate lignin phenols (0.46−0.95; $p < 0.05$; Figure 3d). Both (Ad/Al)$_V$ and (Ad/Al)$_S$ ratios of dissolved lignin phenols, fluctuating along the Shaliu River, were consistently lower in the pre-monsoon than monsoon season ($p < 0.05$; Figure 3d). After examining lignin phenols in the soils of the Shaliu basin, we found spatially variable concentrations of lignin phenols (Figure 4a) but consistently

higher (Ad/Al)$_V$ and (Ad/Al)$_S$ ratios in the top- than subsoil at all sampling sites ($p < 0.05$; Figures 4b-c).

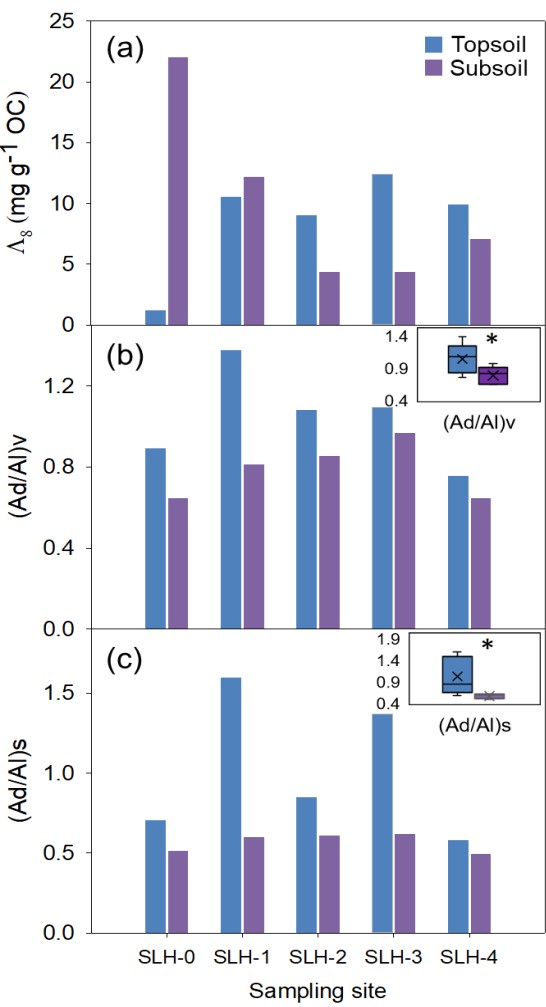

**Figure 4.** Organic carbon (OC)-normalized concentration of lignin phenols ($\Lambda_8$, a), the acid-to-aldehyde (Ad/Al) ratios of vanillyl (V, b) and syringyl (S, c) phenols in the soil of Shaliu River basin. The solid bar and cross in the inserted boxes in panels (b) and (c) mark the median and mean of each dataset, respectively. The upper and lower ends of boxes denote the 0.25

and 0.75 percentiles, respectively. Asterisks indicate significant differences between topsoil (0–10 cm) and subsoil (40–60 cm; paired-sample t tests, n = 5, $p < 0.05$). The legends in panel (a) also apply to other panels.

### 3.3 Carbon variations during local hydrological events

To reveal the riverine carbon variations induced by soil-river water transfer in the Shaliu River, we focused on two important hydrological events. First, we monitored DOC and lignin phenol concentrations in soil solutions along with the
progress of thawing (when soil temperature reached $> 0°C$). Topsoil DOC and lignin phenols showed an increasing (albeit not statistically significant) trends from 19.1 to 22.0 mg $L^{-1}$ and from 44.9 to 57.6 μg $L^{-1}$ on May 11 to June 17 at SLH-1 station, respectively (Figures 5a-c). Similarly, topsoil DOC increased from 22.4 to 29.4 mg $L^{-1}$ on April 22 to June 17 at SLH-3 station (Figure 5b). Subsoil-derived DOM was gradually released with thawing, indicated by the increase of DOC (or lignin phenol) concentration from not detectable (frozen) on May 11 to 13.3 mg $L^{-1}$ (lignin phenols = 23.1 μg $L^{-1}$) on June 17 at SLH-1
station and from not detectable on April 22 to 22.1 mg $L^{-1}$ on May 22 at SLH-3 station, respectively (Figures 5a-c). Furthermore, lignin Ad/Al ratios were lower in the leachates of thawed soils than in the topsoil solution ($p < 0.05$) but similar to the subsoil solution at SLH-1 station (Figures 5d-e), indicating significant contribution of deep soil DOM to the leachate. Concurrently, riverine DOC increased from 2.3 to 3.1 mg $L^{-1}$ on May 11 to June 17 and from 1.9 to 3.9 mg $L^{-1}$ on April 22 to June 17 at SLH-1 and SLH-3 stations (Figures 5a-b). Second, a typically local precipitation event lasting for ~1 hour at SLH-4 station
was monitored to investigate the influence of precipitation on riverine carbon transport. With the start of rain, riverine DOC concentration increased from 2.0 mg $L^{-1}$ to 3.3 mg $L^{-1}$ within 0.5 hour due to flushing and leaching of terrestrially derived OM, and then decreased to 2.1 mg $L^{-1}$ at the end of precipitation event due to dilution by rainwater (Table 2). Simultaneously, POC and PIC concentrations increased from 0.36 to 0.46 mg $L^{-1}$ and from 0.03 to 0.06 mg $L^{-1}$ accompanied by an increase of TSS concentrations from 5.8 to 7.4 mg $L^{-1}$ (Table 2).

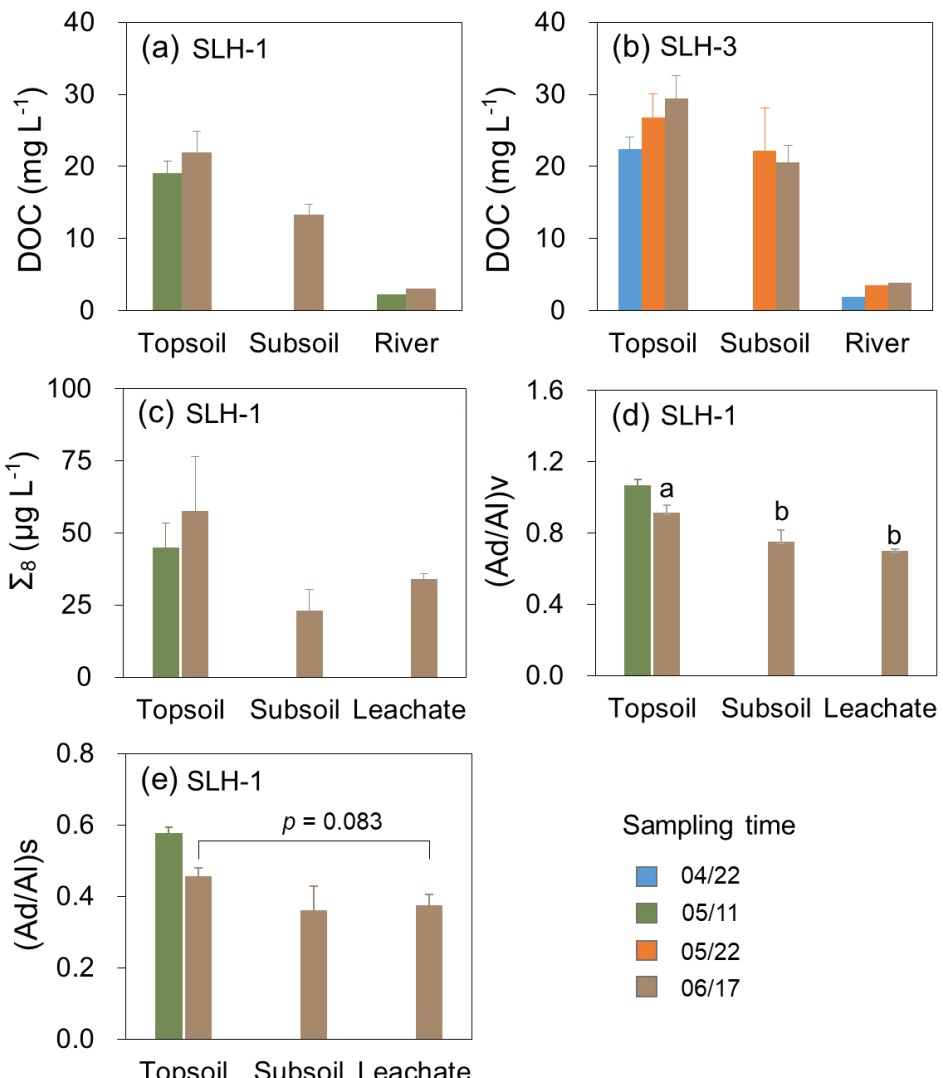


**Figure 5.** Variations of dissolved organic carbon (DOC) in soil solutions and river water at SLH-1 (a) and SLH-3 (b) stations, absolute concentration of lignin phenols ($\Sigma_8$) in soil solutions and leachates in SLH-1 station (c), and the variations of the acid-to-aldehyde (Ad/Al) ratios of syringyl (S) and vanillyl (V) phenols in soil solutions and leachates at SLH-1 station (d-e) during thawing period in 2018. Error bars in panels (a-e) represent standard error of mean (n = 4). Lowercase letters in panel (d)
indicate different levels of Ad/Al ratio among the leachates, top- and subsoil solutions (one-way ANOVA, n = 4, $p < 0.05$) while $p$ values in panel (e) indicate marginal difference of Ad/Al ratio between topsoil solutions and leachates.

**Table 2.** River water properties at the SLH-4 station of Shaliu River during a precipitation event in August 2015.

| Sample | Sampling time | T | pH | Cond. | DO | TSS | DIC | DOC | PIC | POC |
|--------|---------------|---|----|-------|----|-----|-----|-----|-----|-----|
| | (h) | (°C) | | (µs cm$^{-1}$) | | | (mg L$^{-1}$) | | | |
| SLH-4 | 0.25 | 11.4 | 8.2 | 386 | 5.9 | 5.8 | 37.76 | 2.04 | 0.03 | 0.36 |
| SLH-4 | 0.50 | 11.1 | 8.3 | 381 | 5.6 | 6.2 | 37.43 | 3.26 | 0.04 | 0.44 |
| SLH-4 | 0.75 | 11.0 | 8.3 | 384 | 5.5 | 7.0 | 37.24 | 3.21 | 0.04 | 0.42 |
| SLH-4 | 1.00 | 11.5 | 8.3 | 384 | 5.2 | 7.4 | 37.96 | 2.06 | 0.06 | 0.46 |
| **Mean** | | **11.3** | **8.3** | **384** | **5.6** | **6.6** | **37.60** | **2.64** | **0.04** | **0.42** |

T, water temperature; Cond., conductivity; DO, dissolved oxygen; TSS, total suspended solid; DIC, dissolved inorganic carbon; DOC, dissolved organic carbon; PIC, particulate inorganic carbon; POC, particulate organic carbon.

## 4. Discussion

### 4.1 Riverine carbon fluxes in the Shaliu River

In contrast to Arctic rivers with a pronounced spring freshet from May to June, both DOC and DIC fluxes in the Shaliu River were highest from July to August (accounting for 58-60% of the corresponding annual fluxes; Figures 2a-b) due to the maximum discharge caused by frequent precipitation events in the monsoon season (Figure S1; Zeng et al., 2019). DIC, mainly composed of bicarbonate and carbonate based on the river water pH values (Table 1; Liu et al., 2020), was the dominant form of dissolved riverine carbon. Its proportion in dissolved riverine carbon is generally higher than tropical (~40%; Huang et al., 2012) and Arctic rivers (52-70%; Striegl et al., 2007; Prokushkin et al., 2011; Guo et al., 2012) but similar to rivers sourced from the Qinghai-Tibetan Plateau including the upper Yangtze River, Yellow River and their tributaries (Cai et al., 2008; Gao et al., 2019; Song et al., 2019; Song et al., 2020). The distinct lithology (such as limestone and sandstone) within the Shaliu River catchment results in high carbonate and silicate weathering rate (Xiao et al., 2013), which is an important source of DIC in river water. The connection of DIC in the Shaliu river to high chemical weathering rate on the Qinghai-Tibetan Plateau is further reflected by the high riverine $Ca^{2+}$ and $Mg^{2+}$ concentrations (Zhang et al., 2013) and their positive correlations with DIC ($p < 0.05$; Figure S2b). The negative correlation of DIC concentration with river discharge at SLH-4 station (Figure 2a) indicates interactive impacts on hydrology by precipitation and thawing. Specifically, the inputs of subsurface flow and groundwater containing high weathering products (including DIC) under base flow conditions (including during thawing) in this permafrost-affected watershed (Walvoord and Striegl, 2007; Giesler et al., 2014) results in relatively high DIC concentraions. By contrast, riverine DIC concentration decreases with discharge in the monsoon season during 2015 to 2016 likely due to the dilution effects of rainwater, which normally has a lower DIC concentration relative to stream water (Song et al., 2019).

DOC was the second most abundant riverine carbon with a lower concentration in the Shaliu River than most Arctic rivers (Mann et al., 2016; Amon et al., 2012; Spencer et al., 2008), only accounting for 7% of total riverine carbon. However, its concentration is equivalent to the upper Yangtze, Yellow, Lancang and Yarlung Zangbo Rivers (Qu et al., 2017; Ran et al., 2013; Liu et al., 2021). The low DOC concentration reflects the relatively low SOC density in the alpine grasslands of the Qinghai-Tibetan Plateau (9.05 kg C m$^{-2}$ in 0$-$100 cm depth; Yang et al., 2008) compared to most Arctic river basins containing organic matter-rich peatlands and deposits (32.2–69.6 kg C m$^{-2}$; Tarnocai et al., 2009). The DOC/DON ratios in the Shaliu River were generally lower than the global riverine average (14; Harrison et al., 2005; Seitzinger et al., 2005), potentially indicating high biodegradability of riverine DOM (Wiegner et al., 2006), which may contribute to the low DOC concentration in this river as well. Furthermore, DOC concentraions showed no consistent relationship with river discharge throughout the year, while it showed opposite relationships with river discharge in early versus late pre-monsoon seasons (Figure S4). Snowmelt in the early pre-monsoon season (early April) may dilute the base-flow DOC, resulting in an decreasing trend with discharge. By comparison, thawing of frozen soils in late pre-monsoon season (late April to June) releases more frozen carbon and thus results in an increasing trend of DOC with discharge.

## 4.2 Spatial-temporal variation of riverine carbon in the Shaliu River

The concentrations of dissolved and particulate carbon fluctuated along the Shaliu River in the monsoon season while increased significantly downstream in the pre-monsoon. Similarly, the absolute concentration of dissolved and particulate lignin phenols peaked at SLH-2 in the monsoon season, but they showed a downstream increasing trend in the pre-monsoon season. These trends stand in contrast to the downstream accumulation of riverine carbon in the high flow and increasing degradation in the low flow conditions observed in other rivers including Zambezi River (Lambert et al., 2016) and a tributary of Yukon River (Dornblaser and Striegl, 2015), suggesting unique seasonality in the spatial variations of riverine carbon in the Shaliu River. In addition, the higher particulate carbon was directly related to the higher TSS concentrations in the pre-monsoon than monsoon season. Thermal erosion during thawing is the most important pathway supplying particulates and weathering products into the river on the Qinghai-Tibetan Plateau (Wang et al., 2016), which may explain the high particulate concentration in pre-monsoon season. In contrast, other than aged DOC sourced from thawed soils, exudates from plant roots are also an important supply to riverine DOC. Although we did not measure root exudates, we postulate that the higher riverine DOC during the monsoon than pre-monsoon season is related to increased plant growth and exudation in the growing season.

Both (Ad/Al)v and (Ad/Al)s ratios of dissolved lignin phenols were lower in the pre-monsoon season than monsoon season, consistent with the Arctic rivers (Amon et al., 2012) but different from the invariant or opposite patterns between seasons in tropical rivers including Congo and Oubangui (Spencer et al., 2010b; Bouillon et al., 2012). Higher acid-to-aldehyde ratio of lignin phenols is commonly attributed to increased photo- or microbial oxidation of lignin. However, photo- and microbial oxidation may be partially constrained by the short residence time (indicated by high discharge in monsoon season) and low temperature in this alpine stream (average water temperature of 10.8°C in the monsoon season; Table 1). Alternatively, the acid-to-aldehyde (Ad/Al) ratios of lignin phenols in topsoil were higher than those in the subsoil, in contrast to the increase

of Ad/Al ratios with soil depth typically reported in other soils (Otto and Simpson, 2006). Our previous research also finds higher Ad/Al ratios in top- than subsoil in an alpine grassland on the Qinghai-Tibetan Plateau due to the influence of dominant vegetation (i.e., shallow-rooted *K. humilis*) having high Ad/Al ratios in its roots (Jia et al., 2019). Here, the Shaliu River basin is dominated by shallow-rooted *K. humilis* as well (Li et al., 2013), likely leading to the higher Ad/Al ratios in topsoil. Streamflow is supplied by snowmelt water via surface and near-surface pathways in early pre-monsoon season when

temperature ranges from -2 to 0°C (Tetzlaff et al., 2015) and thus contains amounts of carbon from surface soils, while streamflow shifts from surface meltwater to water stored in subsurface frozen soils (i.e., an increasing contribution of deep soil water) as air and soil temperature rises. This transition in streamwater sources in pre-monsoon season suggests that the contribution of deep soil water to streamflow increases with the progress of thawing events (Carey and Quinton, 2004; Tetzlaff et al., 2015). In combination with the different Ad/Al ratios between pre-monsoon and monsoon seasons, we postulate that

topsoil makes a relatively larger contribution to riverine DOM in the monsoon season likely through increased surface runoff, resulting in higher acid-to-aldehyde ratios of dissolved lignin in the stream compared to the pre-monsoon season.

## 4.3 Riverine carbon variations influenced by local hydrological events

The unique spatial-temporal variations of riverine carbon mentioned above are likely related to the high sensitivity of the Shaliu River to local hydrological processes (Biggs et al., 2017), including sporadic occurrence of local precipitation in the

monsoon season (Wu et al., 2016) and thawing of frozen soils in pre-monsoon (spring) season. Specifically, the downstream increase of riverine carbon in the pre-monsoon season (April-May) was anticipated to be related to spring thawing releasing DOC previously frozen in the soil. The observed increase of soil DOC and lignin phenols in topsoil solutions over time reflects carbon release from previously frozen topsoil and/or inputs via lateral flow paths during thawing. Combined with the release of subsoil-derived DOM proved by subsoil DOC availability and lignin phenol Ad/Al ratios, these results indicate the release

of frozen DOM during thawing events. The riverine DOC shows a high synchronousness with the increase of soil DOM, partially explaining the downstream increase of DOC along the Shaliu River in the pre-monsoon season. In particular, as the upstream Shaliu basin has a slightly higher elevation, soil temperature is slightly lower in the upstream than downstream in spring (−1.2~0°C at SLH-1 versus 0~2.9°C at SLH-3 on March 26 to April 20; Figure 1b), leading to earlier and stronger thawing of soil in the lower basin. This spatiotemporal variation of freezing-thawing periods affects riverine carbon transport

in the following two aspects. First, the earlier thawing of frozen soils downstream (at a lower elevation) in the Shaliu basin enhances soil carbon release compared with the upstream basin, likely leading to an increasing carbon concentration along the river continuum (Song et al., 2019; Vonk et al., 2015). Second, the thawing depth of frozen soils increases with time in the pre-monsoon season due to increasing temperature, thus causing an increasing riverine carbon concentration with thawing events (Wang et al., 2017; Song et al., 2019). Overall, inputs of DOC from thawed soil increase downstream in spring, leading

to the downstream increase of riverine carbon in this alpine stream.

In contrast, sporadic local precipitation events occur frequently during the monsoon season, leading to fast hydrological variations reflected by increasing TSS concentrations over time in the 1-h precipitation events (Beel et al., 2018). The rapid

response of riverine DOC and POC to the above hydrological alterations indicates the high sensitivity of riverine carbon to short-term precipitation. According to historical precipitation records (Figure S1), more than 90% of the annual precipitation occurs in monsoon season from June to September, mainly in the form of short-lasting and weak precipitation events (intensity < 5 mm; Wu et al., 2016), thus frequently influencing local soil-stream carbon transfer processes. Along the Shaliu River, DOC concentrations in the monsoon season showed no regular downstream variations, likely related to variable local precipitation events frequently occuring in the Shaliu River basin in the summer (Yao et al., 2013; Wu et al., 2016).

## 5. Conclusions

In conclusion, combining annual flux monitoring and dense spatial sampling along the Shaliu River in two contrasting seasons, we provide a benchmark assessment of riverine carbon variations in an alpine headwater catchment on Qinghai-Tibetan Plateau. We show that DIC constitutes the majority (> 87%) of riverine carbon in this alpine stream, dominated by carbon fluxes in the monsoonal summer. In addition, riverine carbon in the Shaliu River increased downstream in the pre-monsoon season due to increasing contribution of organic matter derived from thawed soils while organic matter inputs induced by sporadic precipitation during the monsoon season led to fluctuating concentrations of riverine carbon in the summer. These results indicate a higher sensitivity of riverine DIC than DOC concentration in the alpine stream to local hydrological events, and DOC source appears to change as well. Given projected climate warming on the Qinghai-Tibetan Plateau (Chen et al., 2013), thawing of permafrost and alterations of precipitation regimes may significantly influence the alpine headwater carbon transport, with critical effects on the biogeochemical cycles of the downstream rivers. On the other hand, the alpine headwater catchments may be utilized as sentinels for climate-induced changes in the hydrological pathways and/or biogeochemistry of the small basin. Both aspects deserve further research attention in the future.

*Data availability*. All data are available within this paper and in the Supplement.

*Supplement*. The supplement related to this article is available online.

*Author contribution*. X.F. designed the study. X.W. performed geochemical and lignin phenol analyses and analyzed related data. T.L. wrote the manuscript. X.W. and T.L. contributed equally to this work. All authors contributed to the field sampling.

*Competing interests*. The authors declare that they have no conflict of interest.

**Acknowledgements**

This study was funded by the Chinese National Key Development Program for Basic Research (2019YFA0607303) and the National Natural Science Foundation of China (42025303; 41973075; 31988102). The authors declare no competing financial interests.

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
