# Peer review of "Spatial-temporal variations in riverine carbon strongly influenced by local hydrological events in an alpine catchment"

_Biogeosciences, 2020_

## Referee Comment (RC1) · Anonymous Referee #1 · 20 Jan 2021

This paper investigates land-freshwater linkages and riverine carbon dynamics in the Shaliu River on the Qinghai-Tibetan Plateau, where there is growing interest to quantify the magnitude and sources of terrestrial carbon mobilized into freshwaters within permafrost-affected watersheds. To achieve this, the authors pursue three objectives: to (1) determine seasonal and annual riverine carbon fluxes; (2) using biomarkers, constrain variability in riverine carbon sources across seasonal shifts in hydroclimate (pre- and post monsoon); and (3) assess precipitation effects on carbon mobilization into rivers during a rainfall event. Potential hydroclimate effects on riverine carbon sources are a particularly interesting component of this study. This paper is suited to Biogeosciences and could help to advance understanding of carbon cycling and

land-freshwater linkages in permafrost-affected terrains. However, considerable revisions are needed to better articulate the key messages of this study and to clarify pertinent methods and environmental effects (e.g. freeze-thaw dynamics) on carbon cycling. The comments below and intended to help improve the clarity and depth of your manuscript.

Major comments

1. This paper aims to present an interesting story, but the combined Results and Discussion section is a main obstacle to the authors clearly articulating (and the reader grasping) data trends and the key messages. I think that separating the Results and Discussion will help this paper to more fully reach its potential. The geochemical analyses are interesting and it would help if they were clearly presented in a separate Results section. Consider structuring your Discussion around the objectives you nicely summarize in L60-66.

2. The effects of spatiotemporal variation in freeze-thaw dynamics on C cycling dynamics should be considered in more detail. For instance, Figure 1 nicely illustrates that "freezing period" and "thawing period" (thaw period defined as soil temperature > 0ËŽC, L103) vary by site and depth in the soil profile (even though the box for "freezing period" suggests it has a strictly-defined time interval, approximately December 10 – March 25). This makes me wonder: How do you account for the spatiotemporal variability in freeze-thaw periods in your interpretations? If the frozen status of soil influences C mobilization into the Shaliu River, then can we presume that variability in timing of soil thaw along your sampling sites and across the watershed would influence the quantity and composition of OM entering streams? Please elaborate on this.

3. The Introduction focuses on headwater streams. While interesting, it would greatly benefit the reader to include more background information on other pertinent components of your study, like DOC and DIC sources in permafrost regions and on the Tibetan Plateau (e.g. Song et al. 2020, DOI 10.1088/1748-9326/ab83ac), what lignin

phenols can reveal about OM composition, how hydrology changes during freeze-thaw cycles, etc. This would help to familiarize the reader with key concepts which are at the foundation of your study.

4. As detailed in my comments below, it appears that reporting of data and statistics is incomplete. Please see my comments below regarding the ANOVA (in minor comments), L302, and Figure 3.

Minor comments

1. Sec 2.3. What is the analytical uncertainty of your TC and POC analyses? Please report this. It would generally useful to know and would also help to assure the reader that your [PIC] values (Table 1) are robust and not within the range of analytical uncertainty for TC or POC.

2. Particulates are interesting and important for considering C species and mobilization in cold regions. Although particulates account for a relatively small proportion of total C (Table 1), it would be interesting to elaborate on trends in particulate C, or at least consider them within the broader perspective of particulate mobilization in permafrost terrains.

3. ANOVA is missing from the summary of statistical analyses you performed (Sec. 2.6). Further, from your Results (Fig. 5d, L269), it seems that you must have done a post-hoc test following the ANOVA to determine which categories differed and to assign the letters indicating this. Please clarify.

Additional comments

L52, L72, L78, L245, etc.: Unlike the active layer, permafrost is not a seasonal phenomenon. Permafrost is defined as ground material remaining at or below 0ËŽC for two or more consecutive years (Muller 1943). Therefore, "seasonally thawed permafrost" should be replaced with "active layer". Further, you nicely demonstrate that increases in riverine C pre-monsoon may be sourced from the active layer, but there is no evidence to support that the OM originates from permafrost (L15-18). Additionally, the correct definition of active layer should be provided early in the manuscript, to provide clarification.

Muller, S.W., 1943. Permafrost or permanently frozen ground and related engineering problems. Special Report, Strategic Engineering Study, Intelligence Branch, Office, Chief of Engineers, no.62, 136 pp. Second printing, 1945, 230 pp. (Reprinted in 1947, J.W. Edwards, Ann Arbor, Michigan, 231 pp.)

L12: What is meant by "divergent carbon transport dynamics"?

L14-16: High discharge facilitated DIC production. What actually caused it? Enhanced chemical weathering of minerals associated with increased precipitation?

L18: As noted above, there is no evidence provided to support your attribution of a permafrost C source.

L61-62: "... annual fluvial carbon fluxes on a monthtly basis... " is a bit unclear. It would be clearer if you change to something like, "... to estimate monthly dissolved and particulate carbon fluxes for one year."

L62: "bulk" as in "bulk concentration"?

L63: "dense" = high temporal resolution?

L70: What is the annual discharge of the Shaliu River? Would be interesting to know.

L86 and Sec. 2.5: Because discharge was measured only at SLH-4, I presume that C fluxes were estimated using the concentration measurements from SLH-4? Please clarify in the Methods and Results.

L104-105: What is a "pre-arranged ceramic head"? Is this a porewater sampling device, like lysimeter?

L165-167: Does the statistical analysis for downstream trend in [DOC] account for

autocorrelation among samples? (see also comment for Figure 3)

L175: Based on your pH values reported in Table S2, it might be worth noting here that DIC was primarily HCO3– + CO32–, rather than CO2.

L181: Units of concentration should be consistent throughout the paper (mg L-1, mmol L-1 in Figure S1b).

L225-227: This text is better suited for a Discussion.

L227-230: Useful rationale for this analysis. It would help the reader if this were earlier, perhaps in the Methods (end of Sec. 2.4).

L232-234: Interesting. This text would fit nicely in a Discussion.

L243-245: Another example of good Discussion material.

L246: "... soil-river carbon transfer inducing riverine carbon variations...". I have no idea what this means! ïĄŁ Please clarify.

L247: By "anticipated" do you mean "hypothesized"? It would be interesting and helpful if you clarified your hypotheses early on in your paper, perhaps at the end of the Introduction.

L249-250: But, this increase was only in topsoil. Was it a significant increase? Would help to clarify.

L252-254: How could thawing of subsoil (active layer thickening) increase DOC in the topsoil? Especially given subsoil [DOC] appears to be lower than topsoil [DOC]? (Fig. 5a,d) Would subsoil DOC not be mobilized downslope as the active layer thaws?

L274-275: How much precipitation fell during this rain event? This would be interesting to know, as rainfall can be generally important for mobilizing sediments and POC (e.g. Beel et al. 2018).

Beel, C. R., Lamoureux, S. F., & Orwin, J. F. (2018). Fluvial response to a period of

hydrometeorological change and landscape disturbance in the Canadian High Arctic. Geophysical Research Letters, 45(19), 10-446.

L302: Data Availability: It does not appear that all data are available within the paper and Supplement. I was interested in exploring the raw data used in LOADEST (Sec. 2.5) to estimate C fluxes, but I could not find this data. Please make it available, as indicated.

Table 1. From L274-275, the time points at which these measurements were made is important. Please include this information.

Figure 2: (a) The terminology here ("... concentrations exported...") could be clearer. In other words, the points show measured concentrations and the lines show modeled concentrations from LOADEST? (b) I think It would be more interesting and useful here if you showed measured fluxes as points and modeled fluxes (from LOADEST) as a line. This would allow the reader to more easily visualize DIC and DOC fluxes and assess model fit. Instantaneous discharge is shown in (a), so I think it would be redundant to include in (b).

Figure 3: (a) Does the statistical analysis for downstream trend in [DOC] account for autocorrelation among samples? (see also comment for L165-167) Trends in geochemistry along the Shaliu river reported in Sec. 3.2 would be more clearly shown if (b) and (c) were plotted as points vs. distance, as in (a). (d) Interesting figure. It would be easier to interpret if the data points and inset boxplot were larger. For instance, I can't tell if there are any subsoil solution data points.

Table S2: This table is as interesting and important as Table 1. It would be useful to include in the main text and also include DOC and POC concentrations, rather than their ratio.

Figure S1: (a) Please indicate the sample size for each boxplot. Interesting that particulate concentrations are higher pre-monsoon, whereas dissolved concentrations are

lower. Why? What does this say about hydrologic effects on C mobilization?

---

## Referee Comment (RC2) · Anonymous Referee #2 · 27 Jan 2021

General comments:

The manuscript investigated the riverine carbon dynamics in an alpine headwater system on the Qinghai-Tibetan Plateau where is less monitored. Ideal methodologies were applied to reach the outlined objectives of this research initiative. The manuscript is generally well written, and the data is properly presented, it is well-suited for the journal Biogeosciences. However, there are some issues, listed below, should be considered.

Major comment:

The water sources of the headwater system could be very complicated in the

permafrost-affected area. It could be the precipitation and also could be the soil pore water as the permafrost thaw. The inputs of those two water sources to the river change with time, and it caused inter-annual changes in the physicochemical characteristics. The manuscript focused on the carbon flux changes influenced by hydrological events, therefore I expect to see more discussion on the interaction effects. In mid-June, Fig. 2 revealed the highest water discharge and the lowest DIC concentration throughout the year, however, the DOC concentration was always stable at around 3 mgL-1. The author has the detailed freezing period and thawing period temperature (Fig. 1), and I think it might be used as a piece of strong evidence to descript the input of permafrost soil pore water. So I encourage the authors to discuss more on the fluctuation of DOC concentration with consideration of the hydrological conditions.

Specific comments:

1) Study area: The Shaliu River is about 110 km, however the plotting scale in the map revealed that the distance between SLH-0 and SLH-6 is less than 3 km. Is there a mistake of the plotting scale?

2) Sampling collection: Why do you choose May and August to represent for pre-monsoon season and monsoon season? Please add some description on the monsoon season.

3) Line 225-235: Why the acid-to-aldehyde ratios of lignin phenols in topsoil are consistently higher than those in the subsoil in this region? Does that mean topsoil undergo higher degradation than subsoil?

4) Line 251-254: the DOC and lignin phenols data in this sentence are VERY hard to compare, please reverse this sentence.

5) Figure 5: I would recommend the author to change the legend into individual colors rather than gradients.

6) Table 1: Have you collected the river discharge data during this precipitation event?

---

## Author Comment (AC1) · 22 Feb 2021

The comment was uploaded in the form of a supplement:
https://bg.copernicus.org/preprints/bg-2020-472/bg-2020-472-AC1-supplement.pdf

---

## Author Comment (AC2) · 22 Feb 2021

**Response to Reviewers of bg-2020-472**

**General response to the Editor**

**Editor's comments:**

Dear Xiaojuan Feng,

We are pleased to inform you that the open discussion of your following BG manuscript was closed:
[…]

No more referee comments and short comments will be accepted. Now the public discussion shall be completed as follows:

You - as the contact author - are requested to individually respond to all referee comments (RCs) by posting final author comments on behalf of all co-authors no later than 24 Feb 2021 (final response phase).

[…]

We sincerely thank the editor and reviewers for their supportive and stimulating comments. Following their suggestions, we have made important changes to our manuscript.

For your convenience, the original comments are listed below in black and our replies follow in blue font. As shown in our detailed responses, we have made every effort to address the concerns of all reviewers. We sincerely hope that our responses have adequately addressed all the comments. Thank you again for your consideration.

**Response to Reviewer #1's comments**

This paper investigates land-freshwater linkages and riverine carbon dynamics in the Shaliu River on the Qinghai-Tibetan Plateau, where there is growing interest to quantify the magnitude and sources of terrestrial carbon mobilized into freshwaters within permafrost-affected watersheds. To achieve this, the authors pursue three objectives: to (1) determine seasonal and annual riverine carbon fluxes; (2) using biomarkers, constrain variability in riverine carbon sources across seasonal shifts in hydroclimate (pre- and post monsoon); and (3) assess precipitation effects on carbon mobilization into rivers during a rainfall event. Potential hydroclimate effects on riverine carbon sources are a particularly interesting component of this study. This paper is suited to Biogeosciences and could help to advance understanding of carbon cycling and land-freshwater linkages in permafrost-affected terrains. However, considerable revisions are needed to better articulate the key messages of this study and to clarify pertinent methods and environmental effects (e.g. freeze-thaw dynamics) on carbon cycling. The comments below and intended to help improve the clarity and depth of your manuscript.

We greatly appreciate the reviewer's positive assessment of our manuscript. We have made substantial changes to our manuscript (details below) and hope that our responses have adequately addressed all the comments.

**Major comments:**

1. This paper aims to present an interesting story, but the combined Results and Discussion section is a main obstacle to the authors clearly articulating (and the reader grasping) data trends and the key messages. I think that separating the Results and Discussion will help this paper to more fully reach its potential. The geochemical analyses are interesting and it would help if they were clearly presented in a separate Results section. Consider structuring your

Discussion around the objectives you nicely summarize in L60-66.

The Results and Discussion section is now separated and re-written. Please refer to the highlighted version for details.

2. The effects of spatiotemporal variation in freeze-thaw dynamics on C cycling dynamics should be considered in more detail. For instance, Figure 1 nicely illustrates that "freezing period" and "thawing period" (thaw period defined as soil temperature > 0°C, L103) vary by site and depth in the soil profile (even though the box for "freezing period" suggests it has a strictly-defined time interval, approximately December 10 – March 25). This makes me wonder: How do you account for the spatiotemporal variability in freeze-thaw periods in your interpretations? If the frozen status of soil influences C mobilization into the Shaliu River, then can we presume that variability in timing of soil thaw along your sampling sites and across the watershed would influence the quantity and composition of OM entering streams? Please elaborate on this.

Good point! The spatiotemporal variation of freezing-thawing periods affects riverine carbon dynamics in the following two aspects. First, the earlier thawing of frozen soils downstream (at a lower elevation) in the Shaliu Basin enhances soil carbon release compared with the upstream basin during the freezing-thawing period, likely leading to an increasing carbon concentration along the river continuum (Song et al., 2019; Vonk et al., 2015). Second, the thawing depth of frozen soils increases with time in the pre-monsoon season due to increasing temperature, thus causing an increasing riverine carbon concentration with thawing events (Wang et al., 2017; Song et al., 2019). The above discussion is added in *Lines 365-370* of the revised manuscript.

References:
Song, C., Wang, G., Mao, T., Chen, X., Huang, K., Sun, X., and Hu, Z.: Importance of active layer freeze-thaw cycles on the riverine dissolved carbon export on the Qinghai-Tibet Plateau permafrost region, PeerJ, 7, e7146, 10.7717/peerj.7146, 2019.
Vonk, J. E., Tank, S. E., Bowden, W. B., Laurion, I., Vincent, W. F., Alekseychik, P., Amyot, M., Billet, M. F., Canario, J., Cory, R. M., Deshpande, B. N., Helbig, M., Jammet, M., Karlsson, J., Larouche, J., MacMillan, G., Rautio, M., Anthony, K. M. W., and Wickland, K. P.: Reviews and syntheses: Effects of permafrost thaw on Arctic aquatic ecosystems, Biogeosciences, 12, 7129-7167, 10.5194/bg-12-7129-2015, 2015.
Wang, G. X., Mao, T. X., Chang, J., Song, C. L., and Huang, K. W.: Processes of runoff generation operating during the spring and autumn seasons in a permafrost catchment on semi-arid plateaus, Journal of Hydrology, 550, 307-317, 10.1016/j.jhydrol.2017.05.020, 2017.

3. The Introduction focuses on headwater streams. While interesting, it would greatly benefit the reader to include more background information on other pertinent components of your study, like DOC and DIC sources in permafrost regions and on the Tibetan Plateau (e.g. Song et al. 2020, DOI 10.1088/1748-9326/ab83ac), what lignin phenols can reveal about OM composition, how hydrology changes during freeze-thaw cycles, etc. This would help to familiarize the reader with key concepts which are at the foundation of your study.

Good point! The relevant background information is added in *Lines 52-81* as follows:

"... *This region is covered by large areas of glaciers, permafrost, and seasonally frozen soils (not underlain by permafrost layers), which are strongly affected by thawing with increasing temperatures in the pre-monsoon spring. Thawing frozen soils and deepening of active layers can strongly affect catchment hydrology, including creating vertical and lateral flow paths, increasing soil filtration, enhancing groundwater-surface water exchange and baseflow (Song et al., 2019; Walvoord and Kurylyk, 2016). In addition, this region has a*

*continental monsoon climate and is hence significantly affected by intense precipitation in the summer monsoon (Zou et al., 2017) ..."*

*"Hydrological alterations induced by both thawing of frozen soils and summer monsoon likely result in unique seasonal patterns in riverine carbon dynamics in the Shaliu River compared to headwater streams in other regions including the Arctic or tropical rivers (Zhang et al., 2013). The sources of fluvially exported carbon, including dissolved organic carbon (DOC) and dissolved inorganic carbon (DIC), includes both recently fixed modern carbon from terrestrial plants and aged carbon preserved in frozen soils within this region. It is reported that more than 50% of the dissolved carbon (including DOC and DIC) in rivers on the Qinghai-Tibetan Plateau is from aged carbon sourced from the active layers of permafrost (Song et al., 2020), while precipitation induced surface runoff is known to enhance modern carbon export from the soil surface (Feng et al. 2015). However, the seasonal transport dynamics of these carbon pools along the Qinghai-Tibetan river continuum are relatively poorly investigated. In particular ..."*

*"... The characterization of lignin phenols, which are unique tracers of terrestrial plant derived OM (Hedges and Mann, 1979) and provide useful information on the oxidation stage of terrestrial OM (indicated by lignin acid-to-aldehyde ratios; Bianchi and Canuel, 2011; Hedges et al., 1988), allows us to investigate the sources and transport of terrestrial organic matter in rivers. Based on these investigations ..."*

4. As detailed in my comments below, it appears that reporting of data and statistics is incomplete. Please see my comments below regarding the ANOVA (in minor comments), L302, and Figure 3.

Data and statistics are added to the Supporting Information as "*Dataset for LOADEST*" and in *Lines 189-190* of the main text as below:

*"Differences in the Ad/Al ratios between soil solutions and leachates at SLH-1 station during thawing events were determined using one-way ANOVA followed by post-hoc test."*

**Minor comments:**

1. Sec 2.3. What is the analytical uncertainty of your TC and POC analyses? Please report this. It would generally useful to know and would also help to assure the reader that your [PIC] values (Table 1) are robust and not within the range of analytical uncertainty for TC or POC.

The analytical precision (standard deviation for repeated measurements of standards) is $\pm$ 0.1% and is added in the text.

2. Particulates are interesting and important for considering C species and mobilization in cold regions. Although particulates account for a relatively small proportion of total C (Table 1), it would be interesting to elaborate on trends in particulate C, or at least consider them within the broader perspective of particulate mobilization in permafrost terrains.

Particulate dynamics are added as below:

*"Simultaneously, POC and PIC concentrations increased from 0.36 to 0.46 mg $L^{-1}$ and from 0.03 to 0.06 mg $L^{-1}$, accompanied by an increase of TSS concentrations from 5.8 to 7.4 mg $L^{-1}$."*

*"In addition, the concentrations of particulate carbon were higher in the pre-monsoon*

*than monsoon season, while dissolved carbon had higher concentrations in the monsoon season (Figure S1). In the pre-monsoon season, thawing of frozen soils potentially increased soil erosion (Olefeldt et al., 2016) likely contributing to the higher TSS and particulate carbon concentrations."*

3. ANOVA is missing from the summary of statistical analyses you performed (Sec.2.6). Further, from your Results (Fig. 5d, L269), it seems that you must have done a post-hoc test following the ANOVA to determine which categories differed and to assign the letters indicating this. Please clarify.

This is now clarified in in the Materials and Methods section.

**Additional comments:**

L52, L72, L78, L245, etc.: Unlike the active layer, permafrost is not a seasonal phenomenon. Permafrost is defined as ground material remaining at or below 0° C for two or more consecutive years (Muller 1943). Therefore, "seasonally thawed permafrost" should be replaced with "active layer". Further, you nicely demonstrate that increases in riverine C pre-monsoon may be sourced from the active layer, but there is no evidence to support that the OM originates from permafrost (L15-18). Additionally, the correct definition of active layer should be provided early in the manuscript, to provide clarification.

Muller, S.W., 1943. Permafrost or permanently frozen ground and related engineering problems. Special Report, Strategic Engineering Study, Intelligence Branch, Office, Chief of Engineers, no.62, 136 pp. Second printing, 1945, 230 pp. (Reprinted in 1947, J.W. Edwards, Ann Arbor, Michigan, 231 pp.)

Thank you for the correction! We realize that our study area, although falling in the Qinghai-Tibetan Plateau permafrost zone, harbors both permafrost and seasonally frozen ground/soils not underlain by permafrost. As we do not have deep ground temperature data to confirm the distribution of permafrost within the basin, we now use "seasonally frozen soils" in the revised test to avoid misunderstanding. This is now explained in the Introduction. Your thorough correction is much appreciated.

L12: What is meant by "divergent carbon transport dynamics"?

We meant that the carbon transport dynamics of headwater streams may be different from large rivers. This is rephrased.

L14-16: High discharge facilitated DIC production. What actually caused it? Enhanced chemical weathering of minerals associated with increased precipitation?

This sentence may be misleading. We have revised this sentence as:

*"We show that riverine carbon fluxes in the Shaliu River was dominated by dissolved inorganic carbon, peaking in the summer partly due to high discharge brought by the monsoon."*

L18: As noted above, there is no evidence provided to support your attribution of a permafrost C source.

The "permafrost" is revised as "frozen soil".

L61-62: ". . . annual fluvial carbon fluxes on a monthtly basis. . . " is a bit unclear. It would be clearer if you change to something like, ". . . to estimate monthly dissolved and particulate carbon fluxes for one year."

Thank you! This sentence is revised accordingly.

L62: "bulk" as in "bulk concentration"?

The "bulk" is revised as "bulk concentration".

L63: "dense" = high temporal resolution?

Here "dense" means high spatial resolution, and we have revised the word "dense" to "a high spatial resolution" to avoid ambiguity.

L70: What is the annual discharge of the Shaliu River? Would be interesting to know.

The annual discharge of 25.4 $m^3$ $s^{-1}$ (Wu et al., 2019) is added.

Reference:

Wu, H., Zhao, G., Li, X.-Y., Wang, Y., He, B., Jiang, Z., Zhang, S., and Sun, W.: Identifying water sources used by alpine riparian plants in a restoration zone on the Qinghai-Tibet Plateau: Evidence from stable isotopes, Science of the Total Environment, 697, 10.1016/j.scitotenv.2019.134092, 2019.

L86 and Sec. 2.5: Because discharge was measured only at SLH-4, I presume that C fluxes were estimated using the concentration measurements from SLH-4? Please clarify in the Methods and Results.

It is clarified in the revised manuscript that carbon fluxes were estimated using data from SLH-4 station.

L104-105: What is a "pre-arranged ceramic head"? Is this a porewater sampling device, like lysimeter?

Yes, the ceramic head is a porewater sampling device like soil pore water suction lysimeter, which is composed of a porous ceramic head connected with a small diameter tube for pulling a vacuum and retrieving the sample. Relevant explanation is added in *Lines 118-120* as below:

*"... using pre-arranged ceramic head (0.2-μm) which is a porewater sampling device composed of a porous ceramic head connected with a small diameter tube for pulling a vacuum and retrieving the sample."*

L165-167: Does the statistical analysis for downstream trend in [DOC] account for autocorrelation among samples? (see also comment for Figure 3)

We tested the autocorrelation of DOC concentrations among samples using Durbin-Watson test, and the results (Durbin-Watson values around 2) indicated no significant autocorrelation (Table R1 below). The clarification is added in Methods section as follows:

*"... between riverine carbon concentration and the distance of sampling sites from SLH-0, where the autocorrelation of riverine carbon concentrations among samples was not significant based on Durbin-Watson test."*

*Table R1. Model Summary of* Durbin-Watson test[b].

| Model | R | R Square | Adjusted R Square | Std. Error of the Estimate | Durbin-Watson |
|-------|-----|----------|-------------------|---------------------------|---------------|
| 1 | .964[a] | .929 | .911 | .13273 | 1.749 |

[a] *Predictors: (Constant).*
[b] *Dependent Variable: DOC*

L175: Based on your pH values reported in Table S2, it might be worth noting here that DIC was primarily $HCO_3^- + CO_3^{2-}$, rather than $CO_2$.

Thank you! Relevant information is added in *Lines 295-296* as below:

*"DIC, mainly composed of bicarbonates and carbonates based on the river water pH values (Table S2), was the dominant form of dissolved riverine carbon."*

L181: Units of concentration should be consistent throughout the paper (mg $L^{-1}$, mmol $L^{-1}$ in Figure S1b).

Revised.

L225-227: This text is better suited for a Discussion.

This text is included in the Discussion section of revised manuscript.

L227-230: Useful rationale for this analysis. It would help the reader if this were earlier, perhaps in the Methods (end of Sec. 2.4).

This rationale is added in the Methods.

*"The acid-to-aldehyde (Ad/Al) ratios of V and S phenols are used to indicate lignin oxidation (Opsahl and Benner, 1995) which typically increase with elevated degradation (Otto and Simpson, 2006) …"*

L232-234: Interesting. This text would fit nicely in a Discussion.

This text is included in the Discussion section.

L243-245: Another example of good Discussion material.

This text is included in the Discussion section.

L246: ". . . soil-river carbon transfer inducing riverine carbon variations. . .". I have no idea what this means! Please clarify.

This sentence is clarified as *"To reveal riverine carbon variations induced by water migration from soil to river in the Shaliu River…"*

L247: By "anticipated" do you mean "hypothesized"? It would be interesting and helpful if you clarified your hypotheses early on in your paper, perhaps at the end of the Introduction.

Thank you! Our hypotheses are added in *Lines 79-81* as below:

*"Based on these investigations, we hypothesize that the release of carbon from frozen soils during thawing events leads to a downstream increase of riverine carbon in the pre-monsoon season while carbon inputs through surface runoff in short-term precipitation events may result in a fluctuation of riverine carbon."*

L249-250: But, this increase was only in topsoil. Was it a significant increase? Would help to clarify.

It was only an increasing trend (not a significant increase). The sentence is clarified as *"Soil DOC and lignin phenols showed an increasing (but not statistically significant) trend from 19.1 to 22.0 mg $L^{-1}$…"*

L252-254: How could thawing of subsoil (active layer thickening) increase DOC in the topsoil? Especially given subsoil [DOC] appears to be lower than topsoil [DOC]? (Fig.5a,d) Would subsoil DOC not be mobilized downslope as the active layer thaws?

Thank you! Our previous phrasing was unclear. The increase of DOC in topsoil solution over time was not caused by thawing of subsoil. The increase was likely caused by carbon release from partially frozen topsoil and/or inputs via lateral flow paths. This sentence is revised to avoid ambiguity:

*"The observed increase of soil DOC and lignin phenols in topsoil solutions over time reflects carbon release from previously frozen topsoil and/or inputs via lateral flow paths with thawing. This combined with the release of subsoil-derived DOM proved by subsoil DOC availability and lignin phenol Ad/Al ratios along with thawing progress indicate the release of frozen DOM during thawing events."*

L274-275: How much precipitation fell during this rain event? This would be interesting to know, as rainfall can be generally important for mobilizing sediments and POC (e.g. Beel et al. 2018).

Beel, C. R., Lamoureux, S. F., & Orwin, J. F. (2018). Fluvial response to a period of hydrometeorological change and landscape disturbance in the Canadian High Arctic. Geophysical Research Letters, 45(19), 10-446.

Unfortunately, we did not monitor the rainfall due to logistical reasons. However, as rainfall normally increases (i.e., accumulates) over time within one rain event, rainfall influences on mobilizing sediments and POC may partially be deduced from the positive correlations of time points (sampling time within the rain event) with TSS concentrations ($p < 0.05$). Relevant illustration is added as follows:

*"In contrast, sporadic local precipitation events occur frequently during the monsoon season, leading to fast hydrological variations reflected by increasing TSS concentrations over time in the 1-h precipitation events (Beel et al., 2018). The rapid response of riverine DOC and POC to the above hydrological alterations indicates the high sensitivity of riverine carbon to short-term precipitation."*

L302: Data Availability: It does not appear that all data are available within the paper and Supplement. I was interested in exploring the raw data used in LOADEST (Sec.2.5) to estimate C fluxes, but I could not find this data. Please make it available, as indicated.

Data is added as *"Dataset for LOADEST"* in the Supporting Information.

Table 1. From L274-275, the time points at which these measurements were made is important. Please include this information.

Time points are added in Table 1.

Figure 2: (a) The terminology here (". . . concentrations exported. . .") could be clearer. In other words, the points show measured concentrations and the lines show modeled concentrations from LOADEST? (b) I think It would be more interesting and useful here if you showed measured fluxes as points and modeled fluxes (from LOADEST) as a line. This would allow the reader to more easily visualize DIC and DOC fluxes and assess model fit. Instantaneous discharge is shown in (a), so I think it would be redundant to include in (b).

Thank you! Figure 2 is revised as below.

[Figure]

***Figure 2.*** *Discharge, dissolved inorganic carbon (DIC) and dissolved organic carbon (DOC)*
*concentrations (a) and fluxes (b) exported from Shaliu River at SLH-4 station during 2015*
*and 2016. The modelled concentrations in (a) and modelled fluxes in (b) are derived from*
*from load estimator (LOADEST). The inserted columns in panel (b) showed the seasonal*
*variations of carbon fluxes classified as follows: spring (May to June), summer (July to*
*September), autumn (October to November), and winter (December to the next April).*

Figure 3: (a) Does the statistical analysis for downstream trend in [DOC] account for
autocorrelation among samples? (see also comment for L165-167) Trends in geochemistry
along the Shaliu river reported in Sec. 3.2 would be more clearly shown if (b) and (c) were
plotted as points vs. distance, as in (a). (d) Interesting figure. It would be easier to interpret if
the data points and inset boxplot were larger. For instance, I can't tell if there are any subsoil
solution data points.

Good point! The autocorrelation of [DOC] among samples are excluded using
Durbin-Watson test (details in our response to comment for L165-167). This explanation is
added in Methods section.

Figure 3 is revised as below:

[Figure]

**Figure 3.** Variations of dissolved organic carbon (DOC) in Shaliu River water (a), absolute concentration of lignin phenols ($\Sigma_8$) in riverine dissolved organic matter (DOM; b) and particulate organic matter (POM; c) during the pre-monsoon and monsoon seasons in 2015, the acid-to-aldehyde (Ad/Al) ratios of syringyl (S) and vanillyl (V) phenols in the riverine DOM, POM, soil solutions and leachates (d). The abscissa in panel (a), (b) and (c) mean the distance of sampling sites from SLH-0. The red lines in panel (a) and (b) correspond to the linear regression of data ($p < 0.05$), and the grey shaded regions in panel (a) and (b) show 95% confidence intervals. The inserted box in (d) is the comparison of $(Ad/Al)_V$ and $(Ad/Al)_S$ ratios of dissolved lignin phenols between pre-monsoon and monsoon seasons, respectively, with asterisks indicating significant differences (independent sample t tests, n = 5, $p < 0.05$). The solid bar and cross in the inserted box mark the median and mean of each data set, respectively. The upper and lower ends of box denote the 0.25 and 0.75 percentiles, respectively. NA, not analyzed.

Table S2: This table is as interesting and important as Table 1. It would be useful to include in the main text and also include DOC and POC concentrations, rather than their ratio.

This table is moved into the main text as Table 1 in our revise manuscript.

Figure S1: (a) Please indicate the sample size for each boxplot. Interesting that particulate concentrations are higher pre-monsoon, whereas dissolved concentrations are lower. Why? What does this say about hydrologic effects on C mobilization?

Sample size for each boxplot is added in the caption of Figure S1a.

The higher particulate carbon was directly related to the higher TSS concentrations in the pre-monsoon than monsoon season. Thermal erosion during thawing is the most important pathway supplying particulates and weathering products into the river on the Qinghai-Tibetan Plateau, which may explain the high particulate concentration in pre-monsoon season. In contrast, other than aged DOC sourced from thawed soils, exudates from plant roots are also an important supply to riverine DOC. Although we did not measure root exudates, we postulate that increased riverine DOC during the monsoon season is related to increased plant growth and exudation in the growing season. Relevant explanation is added in the Discussion section.

**Response to Reviewer #2's comments**

**General comments:**

The manuscript investigated the riverine carbon dynamics in an alpine headwater system on the Qinghai-Tibetan Plateau where is less monitored. Ideal methodologies were applied to reach the outlined objectives of this research initiative. The manuscript is generally well written, and the data is properly presented, it is well-suited for the journal Biogeosciences. However, there are some issues, listed below, should be considered.

Thank you for the positive assessment of our manuscript. We have revised our manuscript and hope that our responses (details below) have addressed all the comments.

**Major comment:**

The water sources of the headwater system could be very complicated in the permafrost-affected area. It could be the precipitation and also could be the soil pore water as the permafrost thaw. The inputs of those two water sources to the river change with time, and it caused inter-annual changes in the physicochemical characteristics. The manuscript focused on the carbon flux changes influenced by hydrological events, therefore I expect to see more discussion on the interaction effects. In mid-June, Fig.2 revealed the highest water discharge and the lowest DIC concentration throughout the year, however, the DOC concentration was always stable at around 3 mgL$^{-1}$. The author has the detailed freezing period and thawing period temperature (Fig. 1), and I think it might be used as a piece of strong evidence to descript the input of permafrost soil pore water. So I encourage the authors to discuss more on the fluctuation of DOC concentration with consideration of the hydrological conditions.

Good suggestion! Discussion on the variations of DIC and DOC concentrations with hydrological conditions is added in main text and Supporting Information as follows:

*"... The negative correlation of DIC concentration with river discharge at SLH-4 station (Figure 2a) indicates interactive impacts on hydrology by precipitation and thawing. Specifically, the inputs of subsurface flow and groundwater containing high weathering products (including DIC) under base flow conditions (including during thawing) in this permafrost-affected watershed (Walvoord and Striegl, 2007; Giesler et al., 2014) results in relatively high DIC concentraions, while the dilution of rainwater leads to a significant decrease of DIC concentration in the monsoon season."*

*"Furthermore, DOC concentraions showed no consistent relationship with river discharge throughout the year, while it shows opposite relationships with river discharge in early versus late pre-monsoon seasons (Figure S3). Snowmelt in the early pre-monsoon season (early April) plays an dilution effect on the base-flow DOC, resulting in an*

*decreasing trend with discharge. By comparison, thawing of frozen soils in late pre-monsoon season (late April to June) releases more frozen carbon (details below) and thus results in an increasing trend of DOC with discharge."*

[Figure]

***Figure S3.*** *Relationships between intra-annual DOC concentrations and river discharge at SLH-4 station in Shaliu River. The red line indicates the variation trend of DOC with discharge in pre-monsoon thawing period. Frozen, pre-monsoon thawing, monsoon and post-monsoon freezing periods are referred as December to March, April to June, July to September, October to November based on measured soil temperatures, respectively.*

**Specific comments:**

1) Study area: The Shaliu River is about 110 km, however the plotting scale in the map revealed that the distance between SLH-0 and SLH-6 is less than 3 km. Is there a mistake of the plotting scale?

Thank you! It is a scaling mistake and is now revised.

2) Sampling collection: Why do you choose May and August to represent for pre-monsoon season and monsoon season? Please add some description on the monsoon season.

The Shaliu River basin is under a continental monsoon climate characterized by warm, humid summer and cold, dry winter. Approximately 90% of the annual precipitation occurs between June and September (Zhang et al., 2013). Moreover, soil temperature is generally below 0 °C from middle October to late April (Figure 1b). Hence, we choose May and August to represent pre-monsoon and monsoon seasons, respectively. Related description is added as follows:

*"The Shaliu River basin is under a continental monsoon climate characterized by warm, humid summers and cold, dry winters (Wang et al., 2018). The mean annual temperature is – 0.5°C within the basin (Li et al., 2013) and mean annual precipitation is 370 mm, ~90% of which occurs in the monsoon season (June to September; Zhang et al., 2013; Wu et al., 2019)."*

References:

Zhang, F., Jin, Z., Li, F., Yu, J., and Xiao, J.: Controls on seasonal variations of silicate weathering and $CO_2$ consumption in, two river catchments on the NE Tibetan Plateau, Journal of Asian Earth Sciences, 62, 547-560, 10.1016/j.jseaes.2012.11.004, 2013.

3) Line 225-235: Why the acid-to-aldehyde ratios of lignin phenols in topsoil are consistently higher than those in the subsoil in this region? Does that mean topsoil undergo higher degradation than subsoil?

Good point! The lignin phenol acid-to-aldehyde ratios (Ad/Al) normally increase with soil depth (Otto and Simpson, 2006). However, our previous research also find higher Ad/Al ratios in top- than subsoil on Qinghai-Tibetan Plateau due to the influence of dominant vegetation (i.e., shallow-rooted *K. humilis*) having high Ad/Al ratios in its roots (Jia et al., 2019). Here, the Shaliu River basin is dominated by shallow-rooted *K. humilis* as well (Li et al., 2013), likely leading to the higher Ad/Al ratios in topsoil. The above explanations are added at *Lines 345-349*.

*"In addition, the acid-to-aldehyde (Ad/Al) ratios of lignin phenols in topsoil were higher than those in the subsoil. The Ad/Al ratios normally increase with soil depth (Otto and Simpson, 2006). However, our previous research find higher Ad/Al ratios in top- than subsoil in an alpine grassland on the Qinghai-Tibetan Plateau due to the influence of dominant vegetation (i.e., shallow-rooted K. humilis) having high Ad/Al ratios in its roots (Jia et al., 2019). Here, the Shaliu River basin is dominated by shallow-rooted K. humilis as well (Li et al., 2013), likely leading to the higher Ad/Al ratios in topsoil."*

References:

Otto, A., and Simpson, M. J.: Evaluation of CuO oxidation parameters for determining the source and stage of lignin degradation in soil, Biogeochemistry, 80, 121-142, 10.1007/s10533-006-9014-x, 2006.

Jia, J., Cao, Z., Liu, C., Zhang, Z., Lin, L., Wang, Y., Haghipour, N., Wacker, L., Bao, H., Dittmar, T., Simpson, M. J., Yang, H., Crowther, T. W., Eglinton, T. I., He, J. S., and Feng, X.: Climate warming alters subsoil but not topsoil carbon dynamics in alpine grassland, Global change biology, 25, 4383-4393, 10.1111/gcb.14823, 2019.

Li, C., Li, X., Yang, T., and Li, Y.: Structure and species diversity of meadow community along the Shaliu River in the Qinghai Lake Basin, Arid Zone Research, 30, 1028-1035, 2013.

4) Line 251-254: the DOC and lignin phenols data in this sentence are VERY hard to compare, please reverse this sentence.

This sentence is re-written: *"Subsoil-derived DOM was gradually released with thawing, indicated by the increase of DOC (or lignin phenol) concentration from not detectable (frozen) on May 11 to 13.3 mg L$^{-1}$ (lignin phenols = 23.1 μg L$^{-1}$) on June 17 at SLH-1 station and from not detectable on April 22 to 22.1 mg L$^{-1}$ on May 22 at SLH-3 station (Figure 5a-c)."*

5) Figure 5: I would recommend the author to change the legend into individual colors rather than gradients.

Figure 5 is revised as below:

[Figure]

6) Table 1: Have you collected the river discharge data during this precipitation event?

This is a good point. River discharge data can reflect the hydrology variations caused by precipitation events. However, it is difficult to obtain these data due to logistical reasons. Alternatively, we show the total suspended solid concentration which may partially indicate hydrology conditions due to its positive correlation with discharge (Meybeck et al., 2003).

Reference:

Meybeck, M., Laroche, L., Durr, H. H., and Syvitski, J. P. M.: Global variability of daily total suspended solids and their fluxes in rivers, Global and Planetary Change, 39, 65-93, 10.1016/s0921-8181(03)00018-3, 2003.

---

## Author Response (AR1)

**Response to Reviewers of bg-2020-472**

**General response to the Editor**

**Associate Editor's comments:**

Dear Authors,

I thank you for providing detailed responses to the comments and suggestions offered by two reviewers.

Both reviewers recognized the importance of your exploratory study in the Shaliu River on the Qinghai-Tibetan Plateau, an understudied region in the study of the global carbon cycle, and the novelty of your findings on the seasonal shift in the hydrologically driven export of soil organic carbon. However, the reviewers also raised several critical issues including articulating key messages and structuring the Results and Discussion sections. While your responses well addressed most reviewer comments, I add additional suggestions below to facilitate your revision. I envisage that the manuscript would require a substantial revision to address all raised issues and suggestions. Therefore, I have to recommend 'reconsider after major revisions' and might need to ask the reviewers to reevaluate the revised manuscript.

[*Detailed comments were shown below…*]

I would like to ask you to make all the changes easily identifiable in a marked-up manuscript based on your point-by-point responses to the comments offered by the reviewers and myself. Please also specify the line numbers of the marked-up manuscript in your responses to comments.

Sincerely,

Ji-Hyung Park

Associate Editor, Biogeosciences

We sincerely thank the editor and reviewers for the supportive and stimulating comments. Based on the suggestions, we have made substantial changes to our manuscript.

For your convenience, the original comments are listed below in black and our replies follow in blue font. As shown in our detailed responses, we have made every effort to address the concerns of all reviewers. We sincerely hope that our responses have adequately addressed all the comments. Thank you again for your consideration.

**Detailed comments:**

- Definition of headwater streams: Headwater streams have often been restricted to the first- to third-order streams in the literature. Please define your use of "headwater streams" and explain how you can consider the main stem and tributaries of the Shaliu River based on specific stream order information for the river system.

According to Lin et al. (2019), stream order information for the Shaliu River is added in *Lines 90-91* and *Figure 1a* as follows:

   *"The Shaliu River is composed of first- to third-order streams (Figure 1a) and hence a typical headwater stream (Lin et al., 2019)."*

[Figure]

**Figure 1a.** *Sampling sites along the Shaliu River (a). The map in panel (a) is processed with ArcGIS 10.0. Stream order information is obtained from Lin et al. (2019).*

Reference:

Lin, P. R., Pan, M., Beck, H. E., Yang, Y., Yamazaki, D., Frasson, R., David, C. H., Durand, M., Pavelsky, T. M., Allen, G. H., Gleason, C. J., and Wood, E. F.: Global reconstruction of naturalized river flows at 2.94 million reaches, Water Resour. Res., 55, 6499-6516, https://doi.org/10.1029/2019wr025287, 2019.

- Study objectives: It appears that you wanted to focus on the unique hydro-biogeochemistry of the Shaliu River as a "headwater" system. On the other hand, you gave less priority in linking your findings (and citing latest publications) to the growing interest in the carbon export from the permafrost of the Qinghai-Tibetan Plateau. Please articulate your primary objectives in the last paragraph of Introduction and the abstract ("To assess these aspects"?).

We do want to illustrate the unique hydro-biogeochemistry of riverine carbon in alpine headwater streams on the Qinghai-Tibetan Plateau (compared to tropical headwaters that are more extensively studied in the literature). The latest research on carbon export from the Qinghai-Tibetan Plateau permafrost and primary objectives of our study are added in *Lines 12-16, Lines 61-66 and Lines 70-71*.

- The hydrology during the thawing period (snowmelts): As your new description of the thawing-induced hydrologic changes nicely explain ("Thawing frozen soils and deepening of active layers can strongly affect catchment hydrology, including creating vertical and lateral flow paths, increasing soil filtration, enhancing groundwater-surface water exchange and baseflow"), "lateral flow paths" would also bring substantial amounts of carbon from the surface soil horizons (this has been observed in many systems including even temperate systems). However, you tend to contrast (in a simplifying way) the dominance of deep paths

during the thawing period and the surface flows during the monsoon storm events. I wondered if the rising water table during thawing or snowmelt events would not flush carbon from the surface horizons rich in carbon. Please provide some discussion of the relative importance of the lateral and vertical flow (or deep vs. shallow) paths during thawing or snowmelt events.

Good point! The rising water table during thawing or snowmelt events might bring amounts of carbon from surface soils into streams. However, many research show that streamflow supplied by surface meltwater mainly occurs at the early pre-monsoon season (temperature between −2°C and 0°C), and streamflow would shift from surface meltwater to water stored in subsurface frozen soils (i.e., an increasing contribution of deep soil waters) as air and soil temperature rises (Figure R1; Carey and Quinton, 2004; Tetzlaff et al., 2015; Wang et al., 2017). This transition in streamwater sources indicates the increasing contribution of deep soil water to streamflow with the progress of thawing events. The above explanations and discussion are added in *Lines 358-363*.

[Figure]

*Figure R1. Conceptual model of the hypothesized change in dominant pathways contributing to stream water dominated by different land cover in response to changes in air temperature. Modified from Tetzlaff et al. (2015).*

References:
Tetzlaff, D., Buttle, J., Carey, S. K., McGuire, K., Laudon, H., and Soulsby, C.: Tracer-based assessment of flow paths, storage and runoff generation in northern catchments: a review, Hydrol. Process., 29, 3475-3490, https://doi.org/10.1002/hyp.10412, 2015.
Carey, S. K. and Quinton, W. L.: Evaluating snowmelt runoff generation in a discontinuous permafrost catchment using stable isotope, hydrochemical and hydrometric data, Nord. Hydrol., 35, 309-324, 2004.
Wang, G. X., Mao, T. X., Chang, J., Song, C. L., and Huang, K. W.: Processes of runoff generation operating during the spring and autumn seasons in a permafrost catchment on semi-arid plateaus, J. Hydrol., 550, 307-317, https://doi.org/10.1016/j.jhydrol.2017.05.020, 2017.

- L274-275 (response to the first reviewer comment): Can't you find precipitation data at a nearby site, like in some weather stations near the Lake Tsinghai? You talked about the carbon export "peaking in the summer due to high discharge brought by the monsoon", but it's really challenging to follow up this without resorting to rainfall data.

Precipitation data near the Qinghai Lake is added as Figure S1 (as below) to display the

potential effects of monsoon (June-September) on rainfall and river discharge. For reviewer #1' comments "L274-275: How much precipitation fell during this rain event?", a rainfall of 1.0 mm on the same sampling date (16 August, 2015) at Gonghe weather station near the Qinghai Lake is extracted and added in *Lines 134-135*.

[Figure]

*Figure S1. Variations of the mean daily and monthly precipitations during recent ten years (2009–2018) at Gonghe weather station near Qinghai Lake (data modified from http://data.cma.cn/data/index/6d1b5efbdcbf9a58.html).*

- Geological information: Given the quantitative importance of DIC, it would help readers if you provide some information (and discussion) on the distribution of bedrocks in the basin.

Good point! Relevant information is added in *Lines 95-96,Lines 311-312* as below:

"*The soils in the basin are mainly Gelic Cambisol (IUSS working group WRB, 2015) underlain by a dominance of Triassic sandstone, late Cambrian metamorphic rocks (schist and gneiss) and granites (Zhang et al., 2013).*"

"*The distinct lithology within the Shaliu River basin results in high carbonate and silicate weathering rate (Zhang et al., 2013), which is an important source of DIC in river water.*"

- Details on sampling and analysis: Given the importance of carbon analysis in this study, you need to provide more details on sampling bottles (glass or plastic? Acid-washed?), SPE, TOC analyzer (high-temperature oxidation? Accuracy?), QC…

Relevant information is added in *Lines 110-112, Lines 160-166, Lines 149-153* in the Materials and Methods section.

**Response to Reviewer #1's comments**

This paper investigates land-freshwater linkages and riverine carbon dynamics in the Shaliu River on the Qinghai-Tibetan Plateau, where there is growing interest to quantify the magnitude and sources of terrestrial carbon mobilized into freshwaters within permafrost-affected watersheds. To achieve this, the authors pursue three objectives: to (1)

determine seasonal and annual riverine carbon fluxes; (2) using biomarkers, constrain variability in riverine carbon sources across seasonal shifts in hydroclimate (pre- and post monsoon); and (3) assess precipitation effects on carbon mobilization into rivers during a rainfall event. Potential hydroclimate effects on riverine carbon sources are a particularly interesting component of this study. This paper is suited to Biogeosciences and could help to advance understanding of carbon cycling and land-freshwater linkages in permafrost-affected terrains. However, considerable revisions are needed to better articulate the key messages of this study and to clarify pertinent methods and environmental effects (e.g. freeze-thaw dynamics) on carbon cycling. The comments below and intended to help improve the clarity and depth of your manuscript.

We greatly appreciate the reviewer's positive assessment of our manuscript. We have made substantial changes to our manuscript (details below) and hope that our responses have adequately addressed all the comments.

**Major comments:**

1. This paper aims to present an interesting story, but the combined Results and Discussion section is a main obstacle to the authors clearly articulating (and the reader grasping) data trends and the key messages. I think that separating the Results and Discussion will help this paper to more fully reach its potential. The geochemical analyses are interesting and it would help if they were clearly presented in a separate Results section. Consider structuring your Discussion around the objectives you nicely summarize in L60-66.

The Results and Discussion section is now separated and re-written. Please refer to the highlighted version for details.

2. The effects of spatiotemporal variation in freeze-thaw dynamics on C cycling dynamics should be considered in more detail. For instance, Figure 1 nicely illustrates that "freezing period" and "thawing period" (thaw period defined as soil temperature > 0°C, L103) vary by site and depth in the soil profile (even though the box for "freezing period" suggests it has a strictly-defined time interval, approximately December 10 – March 25). This makes me wonder: How do you account for the spatiotemporal variability in freeze-thaw periods in your interpretations? If the frozen status of soil influences C mobilization into the Shaliu River, then can we presume that variability in timing of soil thaw along your sampling sites and across the watershed would influence the quantity and composition of OM entering streams? Please elaborate on this.

Good point! The spatiotemporal variation of freezing-thawing periods affects riverine carbon dynamics in the following two aspects. First, the earlier thawing of frozen soils downstream (at a lower elevation) in the Shaliu Basin enhances soil carbon release compared with the upstream basin during the freezing-thawing period, likely leading to an increasing carbon concentration along the river continuum (Song et al., 2019; Vonk et al., 2015). Second, the thawing depth of frozen soils increases with time in the pre-monsoon season due to increasing temperature, thus causing an increasing riverine carbon concentration with thawing events (Wang et al., 2017; Song et al., 2019). The above discussion is added in *Lines 379-384* of the revised manuscript.

References:
Song, C. L., Wang, G., Mao, T. X., Chen, X. P., Huang, K. W., Sun, X. Y., and Hu, Z. Y.: Importance of active layer freeze-thaw cycles on the riverine dissolved carbon export on the Qinghai-Tibet Plateau permafrost region, PeerJ, 7, 25, https://doi.org/10.7717/peerj.7146, 2019.
Vonk, J. E., Tank, S. E., Bowden, W. B., Laurion, I., Vincent, W. F., Alekseychik, P., Amyot, M., Billet, M. F., Canario, J., Cory, R. M., Deshpande, B. N., Helbig, M., Jammet, M., Karlsson, J., Larouche, J., MacMillan, G., Rautio, M., Anthony, K. M. W., and Wickland, K. P.: Reviews and

syntheses: Effects of permafrost thaw on Arctic aquatic ecosystems, Biogeosciences, 12, 7129-7167, https://doi.org/10.5194/bg-12-7129-2015, 2015.

Wang, G. X., Mao, T. X., Chang, J., Song, C. L., and Huang, K. W.: Processes of runoff generation operating during the spring and autumn seasons in a permafrost catchment on semi-arid plateaus, J. Hydrol., 550, 307-317, https://doi.org/10.1016/j.jhydrol.2017.05.020, 2017.

3. The Introduction focuses on headwater streams. While interesting, it would greatly benefit the reader to include more background information on other pertinent components of your study, like DOC and DIC sources in permafrost regions and on the Tibetan Plateau (e.g. Song et al. 2020, DOI 10.1088/1748-9326/ab83ac), what lignin phenols can reveal about OM composition, how hydrology changes during freeze-thaw cycles, etc. This would help to familiarize the reader with key concepts which are at the foundation of your study.

*Good point! The relevant background information is added in Lines 56-61, Lines 61-66, Lines 78-81.*

4. As detailed in my comments below, it appears that reporting of data and statistics is incomplete. Please see my comments below regarding the ANOVA (in minor comments), L302, and Figure 3.

Data and statistics are added to the Supporting Information as "*Dataset for LOADEST*" and in *Lines 201-202* of the main text as below:

*"Differences in the Ad/Al ratios between soil solutions and leachates at SLH-1 station during thawing events were determined using one-way ANOVA followed by post-hoc test."*

**Minor comments:**

1. Sec 2.3. What is the analytical uncertainty of your TC and POC analyses? Please report this. It would generally useful to know and would also help to assure the reader that your [PIC] values (Table 1) are robust and not within the range of analytical uncertainty for TC or POC.

The analytical precision (standard deviation for repeated measurements of standards) is $\pm$ 0.1% and is added in the text.

2. Particulates are interesting and important for considering C species and mobilization in cold regions. Although particulates account for a relatively small proportion of total C (Table 1), it would be interesting to elaborate on trends in particulate C, or at least consider them within the broader perspective of particulate mobilization in permafrost terrains.

Particulate dynamics are added in *Lines 287-288, Lines 340-343*.

3. ANOVA is missing from the summary of statistical analyses you performed (Sec.2.6). Further, from your Results (Fig. 5d, L269), it seems that you must have done a post-hoc test following the ANOVA to determine which categories differed and to assign the letters indicating this. Please clarify.

This is now clarified in the Materials and Methods section.

**Additional comments:**

L52, L72, L78, L245, etc.: Unlike the active layer, permafrost is not a seasonal phenomenon. Permafrost is defined as ground material remaining at or below 0° C for two or more consecutive years (Muller 1943). Therefore, "seasonally thawed permafrost" should be replaced with "active layer". Further, you nicely demonstrate that increases in riverine C pre-monsoon may be sourced from the active layer, but there is no evidence to support that the OM originates from permafrost (L15-18). Additionally, the correct definition of active

layer should be provided early in the manuscript, to provide clarification.

Muller, S.W., 1943. Permafrost or permanently frozen ground and related engineering problems. Special Report, Strategic Engineering Study, Intelligence Branch, Office, Chief of Engineers, no.62, 136 pp. Second printing, 1945, 230 pp. (Reprinted in 1947, J.W. Edwards, Ann Arbor, Michigan, 231 pp.)

Thank you for the correction! We realize that our study area, although falling in the Qinghai-Tibetan Plateau permafrost zone, harbors both permafrost and seasonally frozen ground/soils not underlain by permafrost. As we do not have deep ground temperature data to confirm the distribution of permafrost within the basin, we now use "seasonally frozen soils" in the revised text to avoid misunderstanding. This is now explained in the Introduction. Your thorough correction is much appreciated.

L12: What is meant by "divergent carbon transport dynamics"?

We meant that the carbon transport dynamics of headwater streams may be different from large rivers. This is rephrased.

L14-16: High discharge facilitated DIC production. What actually caused it? Enhanced chemical weathering of minerals associated with increased precipitation?

This sentence may be misleading. We have revised this sentence in *Lines 18-19 as below*:

*"We show that riverine carbon fluxes in the Shaliu River was dominated by dissolved inorganic carbon, peaking in the summer partly due to high discharge brought by the monsoon."*

L18: As noted above, there is no evidence provided to support your attribution of a permafrost C source.

The "permafrost" is revised as "frozen soils" in *Line 22*.

L61-62: ". . . annual fluvial carbon fluxes on a monthtly basis. . . " is a bit unclear. It would be clearer if you change to something like, ". . . to estimate monthly dissolved and particulate carbon fluxes for one year."

Thank you! This sentence is revised accordingly in *Lines 73-74*.

L62: "bulk" as in "bulk concentration"?

The "bulk" is revised as "bulk concentration" in *Line 74*.

L63: "dense" = high temporal resolution?

Here "dense" means high spatial resolution, and we have revised the word "dense" to "a high spatial resolution" to avoid ambiguity in *Line 75*.

L70: What is the annual discharge of the Shaliu River? Would be interesting to know.

The annual discharge of 25.4 m$^3$ s$^{-1}$ (Wu et al., 2019) is added in *Lines 88-89*.

Reference:

Wu, H. W., Zhao, G. Q., Li, X. Y., Wang, Y., He, B., Jiang, Z. Y., Zhang, S. Y., and Sun, W.: Identifying water sources used by alpine riparian plants in a restoration zone on the Qinghai-Tibet Plateau: Evidence from stable isotopes, Sci. Total Environ., 697, 11, https://doi.org/10.1016/j.scitotenv.2019.134092, 2019.

L86 and Sec. 2.5: Because discharge was measured only at SLH-4, I presume that C fluxes were estimated using the concentration measurements from SLH-4? Please clarify in the Methods and Results.

It is clarified in the revised manuscript that carbon fluxes were estimated using data from SLH-4 station in *Line 186, Line 211 and Line 214*.

L104-105: What is a "pre-arranged ceramic head"? Is this a porewater sampling device, like lysimeter?

Yes, the ceramic head is a porewater sampling device like soil pore water suction lysimeter, which is composed of a porous ceramic head connected with a small diameter tube for pulling a vacuum and retrieving the sample. Relevant explanation is added in *Lines 129-130*.

L165-167: Does the statistical analysis for downstream trend in [DOC] account for autocorrelation among samples? (see also comment for Figure 3)

We tested the autocorrelation of DOC concentrations among samples using Durbin-Watson test, and the results (Durbin-Watson values around 2) indicated no significant autocorrelation (Table R1 below). The clarification is added in Methods section in *Lines 204-205*.

*Table R1. Model Summary of Durbin-Watson test[b].*

| Model | R | R Square | Adjusted R Square | Std. Error of the Estimate | Durbin-Watson |
|-------|-----|----------|-------------------|----------------------------|---------------|
| 1 | .964[a] | .929 | .911 | .13273 | 1.749 |

[a] *Predictors: (Constant).*
[b] *Dependent Variable: DOC*

L175: Based on your pH values reported in Table S2, it might be worth noting here that DIC was primarily $HCO_3^- + CO_3^{2-}$, rather than $CO_2$.

Thank you! Relevant information is added in *Lines 306-307*.

L181: Units of concentration should be consistent throughout the paper (mg $L^{-1}$, mmol $L^{-1}$ in Figure S1b).

The Figure S1b is revised as Figure S2b.

L225-227: This text is better suited for a Discussion.

This text is included in the Discussion section of revised manuscript in *Lines 347-358*.

L227-230: Useful rationale for this analysis. It would help the reader if this were earlier, perhaps in the Methods (end of Sec. 2.4).

This rationale is added in the Methods in *Lines 182-183*.

L232-234: Interesting. This text would fit nicely in a Discussion.

This text is included in the Discussion section in *Lines 353-358.*

L243-245: Another example of good Discussion material.

This text is included in the Discussion section 4.3.

L246: ". . . soil-river carbon transfer inducing riverine carbon variations. . .". I have no idea what this means! Please clarify.

This sentence is clarified in *Line 272* as *"To reveal the riverine carbon variations induced by soil-river water transfer in the Shaliu River…"*

L247: By "anticipated" do you mean "hypothesized"? It would be interesting and helpful if you clarified your hypotheses early on in your paper, perhaps at the end of the Introduction.

Thank you! Our hypotheses are added in *Lines 81-84*.

L249-250: But, this increase was only in topsoil. Was it a significant increase? Would help to clarify.

*It was only an increasing trend (not a significant increase). The sentence is clarified in Lines 274-275 as "Topsoil DOC and lignin phenols showed an increasing (albeit not statistically significant) trends from 19.1 to 22.0 mg $L^{-1}$…"*

L252-254: How could thawing of subsoil (active layer thickening) increase DOC in the topsoil? Especially given subsoil [DOC] appears to be lower than topsoil [DOC]? (Fig.5a,d) Would subsoil DOC not be mobilized downslope as the active layer thaws?

*Thank you! Our previous phrasing was unclear. The increase of DOC in topsoil solution over time was not caused by thawing of subsoil. The increase was likely caused by carbon release from partially frozen topsoil and/or inputs via lateral flow paths. This sentence is revised to avoid ambiguity in Lines 372-375.*

L274-275: How much precipitation fell during this rain event? This would be interesting to know, as rainfall can be generally important for mobilizing sediments and POC (e.g. Beel et al. 2018).

    Beel, C. R., Lamoureux, S. F., & Orwin, J. F. (2018). Fluvial response to a period of hydrometeorological change and landscape disturbance in the Canadian High Arctic. Geophysical Research Letters, 45(19), 10-446.

*Unfortunately, we did not monitor the rainfall during this short-term precipitation event due to logistical reasons. However, rainfall on the same day (16 August, 2015) at Gonghe weather station (near Shaliu River and Qinghai Lake) is added to show the potential rainfall within this basin. In addition, as rainfall normally increases (i.e., accumulates) over time within one rain event, rainfall influences on mobilizing sediments and POC may partially be deduced from the positive correlations of time points (sampling time within the rain event) with TSS concentrations ($p < 0.05$). Relevant information and illustration are added in Lines 134-135, Lines 387-389.*

L302: Data Availability: It does not appear that all data are available within the paper and Supplement. I was interested in exploring the raw data used in LOADEST (Sec.2.5) to estimate C fluxes, but I could not find this data. Please make it available, as indicated.

*Data is added as "Dataset for LOADEST" in the Supporting Information.*

Table 1. From L274-275, the time points at which these measurements were made is important. Please include this information.

*Time points are added in Table 2 of the revised manuscript.*

Figure 2: (a) The terminology here (". . . concentrations exported. . .") could be clearer. In other words, the points show measured concentrations and the lines show modeled concentrations from LOADEST? (b) I think It would be more interesting and useful here if you showed measured fluxes as points and modeled fluxes (from LOADEST) as a line. This would allow the reader to more easily visualize DIC and DOC fluxes and assess model fit. Instantaneous discharge is shown in (a), so I think it would be redundant to include in (b).

*Thank you! Figure 2 is revised as below.*

[Figure]

***Figure 2.*** *Discharge, dissolved inorganic carbon (DIC) and dissolved organic carbon (DOC) concentrations (a) and fluxes (b) exported from Shaliu River at SLH-4 station during 2015 and 2016. The modelled concentrations in (a) and modelled fluxes in (b) are derived from load estimator (LOADEST). The inserted columns in panel (b) show the seasonal variations of carbon fluxes classified as follows: spring (May to June), summer (July to September), autumn (October to November), and winter (December to the next April).*

Figure 3: (a) Does the statistical analysis for downstream trend in [DOC] account for autocorrelation among samples? (see also comment for L165-167) Trends in geochemistry along the Shaliu river reported in Sec. 3.2 would be more clearly shown if (b) and (c) were plotted as points vs. distance, as in (a). (d) Interesting figure. It would be easier to interpret if the data points and inset boxplot were larger. For instance, I can't tell if there are any subsoil solution data points.

Good point! The autocorrelation of [DOC] among samples are excluded using Durbin-Watson test (details in our response to comment for L165-167). This explanation is added in Methods section.

Figure 3 is revised as below:

[Figure]

**Figure 3.** Variations of dissolved organic carbon (DOC) in Shaliu River water (a), absolute concentration of lignin phenols ($\Sigma_8$) in riverine dissolved organic matter (DOM; b) and particulate organic matter (POM; c) during the pre-monsoon and monsoon seasons in 2015, the acid-to-aldehyde (Ad/Al) ratios of syringyl (S) and vanillyl (V) phenols in the riverine DOM, POM, soil solutions and leachates (d). The abscissa in panel (a), (b) and (c) mean the distance of sampling sites from SLH-0. The red lines in panel (a) and (b) correspond to the linear regression of data ($p < 0.05$), and the grey shaded regions in panel (a) and (b) show 95% confidence intervals. The inserted box in (d) is the comparison of $(Ad/Al)_V$ and $(Ad/Al)_S$ ratios of dissolved lignin phenols between pre-monsoon and monsoon seasons, respectively, with asterisks indicating significant differences (independent sample t tests, n = 5, $p < 0.05$). The solid bar and cross in the inserted box mark the median and mean of each data set, respectively. The upper and lower ends of box denote the 0.25 and 0.75 percentiles, respectively.

Table S2: This table is as interesting and important as Table 1. It would be useful to include in the main text and also include DOC and POC concentrations, rather than their ratio.

This table is moved into the main text as Table 1 in our revised manuscript.

Figure S1: (a) Please indicate the sample size for each boxplot. Interesting that particulate concentrations are higher pre-monsoon, whereas dissolved concentrations are lower. Why? What does this say about hydrologic effects on C mobilization?

Sample size for each boxplot is added in the caption of Figure S2a.

The higher particulate carbon was directly related to the higher TSS concentrations in the pre-monsoon than monsoon season. Thermal erosion during thawing is the most important pathway supplying particulates and weathering products into the river on the

Qinghai-Tibetan Plateau, which may explain the high particulate concentration in pre-monsoon season. In contrast, other than aged DOC sourced from thawed soils, exudates from plant roots are also an important supply to riverine DOC. Although we did not measure root exudates, we postulate that the higher riverine DOC during the monsoon season is related to increased plant growth and exudation in the growing season. Relevant explanation is added in the Discussion section in *Lines 340-346.*

**Response to Reviewer #2's comments**

**General comments:**

The manuscript investigated the riverine carbon dynamics in an alpine headwater system on the Qinghai-Tibetan Plateau where is less monitored. Ideal methodologies were applied to reach the outlined objectives of this research initiative. The manuscript is generally well written, and the data is properly presented, it is well-suited for the journal Biogeosciences. However, there are some issues, listed below, should be considered.

Thank you for the positive assessment of our manuscript. We have revised our manuscript and hope that our responses (details below) have addressed all the comments.

**Major comment:**

The water sources of the headwater system could be very complicated in the permafrost-affected area. It could be the precipitation and also could be the soil pore water as the permafrost thaw. The inputs of those two water sources to the river change with time, and it caused inter-annual changes in the physicochemical characteristics. The manuscript focused on the carbon flux changes influenced by hydrological events, therefore I expect to see more discussion on the interaction effects. In mid-June, Fig.2 revealed the highest water discharge and the lowest DIC concentration throughout the year, however, the DOC concentration was always stable at around 3 mgL$^{-1}$. The author has the detailed freezing period and thawing period temperature (Fig. 1), and I think it might be used as a piece of strong evidence to descript the input of permafrost soil pore water. So I encourage the authors to discuss more on the fluctuation of DOC concentration with consideration of the hydrological conditions.

Good suggestion! Discussion on the variations of DIC and DOC concentrations with hydrological conditions is added in *Lines 315-319 and Lines 328-332* in main text and Supporting Information (Figure S4 below).

[Figure]

***Figure S4.*** *Relationships between intra-annual DOC concentrations and river discharge at*

*SLH-4 station in Shaliu River. The red line indicates the variation trend of DOC with discharge in pre-monsoon thawing period. Frozen, pre-monsoon thawing, monsoon and post-monsoon freezing periods are referred as December to March, April to June, July to September, October to November based on measured soil temperatures, respectively.*

**Specific comments:**

1) Study area: The Shaliu River is about 110 km, however the plotting scale in the map revealed that the distance between SLH-0 and SLH-6 is less than 3 km. Is there a mistake of the plotting scale?

Thank you! It is a scaling mistake and is now revised in Figure 1a.

2) Sampling collection: Why do you choose May and August to represent for pre-monsoon season and monsoon season? Please add some description on the monsoon season.

The Shaliu River basin is under a continental monsoon climate characterized by warm, humid summer and cold, dry winter. Approximately 90% of the annual precipitation occurs between June and September (Figure S1; Zhang et al., 2013). Moreover, soil temperature is generally below 0 °C from middle October to late April (Figure 1b). Hence, we choose May and August to represent pre-monsoon and monsoon seasons, respectively. Related description is added in *Lines 96-97*.

References:

Zhang, F., Jin, Z. D., Li, F. C., Yu, J. M., and Xiao, J.: Controls on seasonal variations of silicate weathering and $CO_2$ consumption in, two river catchments on the NE Tibetan Plateau, J. Asian Earth Sci., 62, 547-560, https://doi.org/10.1016/j.jseaes.2012.11.004, 2013.

3) Line 225-235: Why the acid-to-aldehyde ratios of lignin phenols in topsoil are consistently higher than those in the subsoil in this region? Does that mean topsoil undergo higher degradation than subsoil?

Good point! The lignin phenol acid-to-aldehyde ratios (Ad/Al) normally increase with soil depth (Otto and Simpson, 2006). However, our previous research also find higher Ad/Al ratios in top- than subsoil on Qinghai-Tibetan Plateau due to the influence of dominant vegetation (i.e., shallow-rooted *K. humilis*) having high Ad/Al ratios in its roots (Jia et al., 2019). Here, the Shaliu River basin is dominated by shallow-rooted *K. humilis* as well (Li et al., 2013), likely leading to the higher Ad/Al ratios in topsoil. The above explanations are added at *Lines 353-358*.

References:

Otto, A. and Simpson, M. J.: Evaluation of CuO oxidation parameters for determining the source and stage of lignin degradation in soil, Biogeochemistry, 80, 121-142, https://doi.org/10.1007/s10533-006-9014-x, 2006.

Jia, J., Cao, Z. J., Liu, C. Z., Zhang, Z. H., Lin, L., Wang, Y. Y., Haghipour, N., Wacker, L., Bao, H. Y., Dittmar, T., Simpson, M. J., Yang, H., Crowther, T. W., Eglinton, T. I., He, J. S., and Feng, X. J.: Climate warming alters subsoil but not topsoil carbon dynamics in alpine grassland, Glob. Change Biol., 25, 4383-4393, https://doi.org/10.1111/gcb.14823, 2019.

Li, C., Li, X., Yang, T., and Li, Y.: Structure and Species Diversity of Meadow Community along the Shaliu River in the Qinghai Lake Basin, Arid Zone Research, 30, 1028-1035, 2013.

4) Line 251-254: the DOC and lignin phenols data in this sentence are VERY hard to compare, please reverse this sentence.

This sentence is re-written in *Lines 277-279*: "*Subsoil-derived DOM was gradually released with thawing, indicated by the increase of DOC (or lignin phenol) concentration from not*

*detectable (frozen) on May 11 to 13.3 mg L$^{-1}$ (lignin phenols = 23.1 µg L$^{-1}$) on June 17 at SLH-1 station and from not detectable on April 22 to 22.1 mg L$^{-1}$ on May 22 at SLH-3 station (Figures 5a-c)."*

5) Figure 5: I would recommend the author to change the legend into individual colors rather than gradients.

Figure 5 is revised as below:

[Figure]

6) Table 1: Have you collected the river discharge data during this precipitation event?

This is a good point. River discharge data can reflect the hydrology variations caused by precipitation events. However, it is difficult to obtain these data due to logistical reasons. Alternatively, we show the total suspended solid concentration which may partially indicate hydrology conditions due to its positive correlation with discharge (Meybeck et al., 2003).

Reference:

Meybeck, M., Laroche, L., Durr, H. H., and Syvitski, J. P. M.: Global variability of daily total suspended solids and their fluxes in rivers, Global and Planetary Change, 39, 65-93, 10.1016/s0921-8181(03)00018-3, 2003.

---

## Referee Report (RR1)

The revised manuscript is much improved. Well done by the authors. While it could be on track for publication, there are a few important considerations and perhaps one methodological error that must first be addressed. Please see my comments below, which are intended to be positive and constructive. I would be happy to review a revised version of the manuscript and I again commend the authors for the great progress they have made with this manuscript.

**Major comments**

1. What were the units for discharge when you modeled constituent fluxes with LOADEST? From your SI dataset, it seems that you used $m^3$ $s^{-1}$? As detailed in Runkel et al. (2004), LOADEST takes discharge in units of $ft^3$ $s^{-1}$. If you used $m^3$ $s^{-1}$, then your constituent modeling must be updated and associated values throughout the text corrected (e.g., Sec. 3.1). Of course, this would be relatively easy to do! This being said, I agree with reporting discharge in metric units ($m^3$ $s^{-1}$) in the main text (e.g., L210), as is commonly done.

   Runkel, R. L., Crawford, C. G., & Cohn, T. A. (2004). *Load Estimator (LOADEST): A FORTRAN program for estimating constituent loads in streams and rivers* (No. 4-A5).
   https://pubs.usgs.gov/tm/2005/tm4A5/pdf/508final.pdf

2. Following on Dr. Park's comment about headwater streams: Headwaters are generally the smallest surficial fluvial component within a stream network. I read Meyer and Wallace (2001) and could not find where they explicitly state that headwaters include $1^{st}$ to $3^{rd}$ order streams, as your citation in L34 implies… but please feel free to prove me wrong about this :) ! The lowest stream order is not necessarily a headwater stream, because definitions of stream order are relative and can vary depending on the resolution of the data used to determine stream order. For instance, stream order derived from geospatial data like a DEM (often relatively coarse, e.g., 90 m from Lin et al., 2019 WRR) may be quite different than stream order determined by an expert in the field or using high-resolution satellite imagery to visually map streams. From Google Earth imagery and your Figure 1, sampling points SLH2–5 do not appear to be headwater streams, and I am not convinced that SLH0–1 are, either. All this to say, I am not convinced that the streams in your study are truly headwater streams and I would not use GIS-derived estimates of stream order (or the citation to Meyer and Wallace 2001) to try to assert this. I think it would be appropriate to place less emphasis on interpreting your results as direct measurements of headwater streams. I think this does not require a major rewrite of your paper and can be addressed without too much trouble by modifying the very nice text that you have in the Introduction and elsewhere. For instance, you could reason that your sampling points capture the integrated hydrochemical signal across headwater catchments. This would be compelling and more accurate. Clarifying this, and omitting 'headwater' where appropriate, would more accurately represent your study sites without comprising the overall relevance of your research, or its significance for understanding hydrochemical signals within headwaters of the QTP.

**Minor comments**

1. Also related to LOADEST: I see now (from the text in Sec. 2.5 and Table S1) that you consider all possible LOADEST regression equations when modeling C flux. i.e., It appears that you selected Model 0 and let LOADEST automatically choose the most parsimonious model

(lowest AIC) from models 1–9. However, model #s 3, 5, 7–9 include covariates for long-term change, which are arguably more appropriate to use when you have measurements from across multiple years (even five years of observations is not considered long enough by some studies, e.g., Zolkos et al., 2020). Because your study period is not multi-annual, and you are not investigating long-term trends, it would be more appropriate to estimate fluxes using the best model from 1, 2, 4, or 6.

Zolkos, S., Krabbenhoft, D. P., Suslova, A., Tank, S. E., McClelland, J. W., Spencer, R. G., ... & Holmes, R. M. (2020). Mercury Export from Arctic Great Rivers. *Environmental Science & Technology*, *54*(7), 4140-4148.

**Additional comments**

L14: "dynamics" is vague. Avoiding it generally improves clarity. If there is more to the story than just C 'transport' / fluxes, consider clarifying how 'dynamics' relates to it.

L17: What do you mean by "in-depth"? Please clarify.

L22: "… thawed frozen…". Are the soils thawed or frozen? I see what you are trying to say here. It would be clearer to just say 'thawed'.

L30: "dynamics"- see comment for L14.

L30: What exactly do you mean by "cascading effects"? i.e., that thaw effects in headwaters might propagate downstream, across watershed scales? (especially for dissolved inorganic carbon, in your region) If so, you might consider recent work from the western Canadian Arctic exploring this topic: https://bg.copernicus.org/articles/17/5163/2020/bg-17-5163-2020.html, but also recognize that thaw effects on fluvial C cycling (especially DIC) across watershed scales vary across permafrost regions. For Siberia, see:
https://agupubs.onlinelibrary.wiley.com/doi/full/10.1029/2017JG004311
Perhaps worth considering in the Discussion, and how your work adds to this understanding.

L63-4: Why is it important that there are different C sources? It would be interesting and helpful to clarify the biogeochemical relevance of the C sources, and doing so could help to justify the rationale for your Ad/Al ratio measurements. For instance, do the sources have varying degrees of recalcitrance?

L89: The value for mean annual daily discharge is interesting. It would also be interesting to know total annual discharge (e.g., $km^3 \, y^{-1}$), from your daily $Q$ measurements.

L90-1: Please see my Major comment #2.

L95-6: No carbonate lithologies? Why are your DIC concentrations so high? This should be discussed a bit more. My recollection is that some of the recent work by Song et al. discusses carbonate lithologies on the QTP.

L202: Please specify: What post-hoc test did you use?

L239: Would be good to consistently include valencies for calcium and magnesium throughout, as you do here (e.g., L156).

L307: Pardon the editorial comment, but it would be more appropriate for dissolved 'bicarbonates' and 'carbonates' to be singular, not plural :)

L311-12: Please see my comment for L95-6. Are there carbonate lithologies? Consistency in the message would help.

L313: Technically, high $Ca^{2+}$ and $Mg^{2+}$ concentrations and a strong association with DIC reflect greater availability of carbonate-bearing lithologies for chemical weathering, not necessarily high rates of weathering. So, I am not sure that I would say "proved" here. Perhaps say "reflected" or "indicated by".

L318-320: Rainfall should enhance carbonate weathering, no? (e.g., Zeng et al., 2019) Perhaps clarify and elaborate on this.

Zeng, S., Liu, Z., & Kaufmann, G. (2019). Sensitivity of the global carbonate weathering carbon-sink flux to climate and land-use changes. *Nature Communications*, *10*(1), 1-10.

L352: "headstream"- i.e., "headwater stream"?

L398: You could simply and clarify by saying "DIC" instead of "dissolved carbon (especially DIC)".

L401: Suggest deleting "frozen" (related to my comment for L22).

L402: "levels" = "concentrations"?

L402: Apparently higher sensitivity of DIC concentration than DOC concentration, but also very interesting that DOC source appears to change. Worth mentioning here.

Figure 2: Are the *y*-axis units metric 'tonnes' per day, rather than imperial 'tons'? For easier comparison with other figures and values, it would help state the *y*-axis in metric units (e.g. Megagrams, Mg, or in grams if desired) and re-scale as needed.

---

## Author Response (AR2)

**Response to Reviewers of bg-2020-472**

**General response to the Editor**

**Associate Editor's comments:**

Dear Authors,

I thank you for your comprehensive revision integrating all the comments and suggestions from the first review round.

A reviewer involved in the first review round has kindly provided another set of very considerate and constructive comments. This reviewer, like myself, thought that your careful revision would further improve the manuscript while reducing some lingering uncertainties that are detailed in the attached reviewer report. Regarding the issue of the term "headwater streams", your perusal of the hydrology literature, in addition to the reviewer comment, would fix the nonsense in sentences like "The Shaliu River is composed of first- to third-order streams (Figure 1a) and hence a typical headwater stream": in other words, a headwater stream consists of headwater streams.

As you did well during the first revision, please make all the changes easily identifiable in a marked-up manuscript and specify the line numbers of the marked-up manuscript in your responses to the reviewer comments.

Sincerely,

Ji-Hyung Park

Associate Editor, Biogeosciences

We sincerely thank the editor and reviewers for the considerate and constructive comments. Following their suggestions, we have made important changes to our manuscript, including the definition of headwater stream and the description of our study site when referring to headwater stream. Please refer to the track change version of manuscript for details.

For your convenience, the original comments are listed below in black and our replies follow in blue font. Line numbers listed below corresponds to the revised version with track changes. As shown in our detailed responses, we have made every effort to address the concerns of all reviewers. We sincerely hope that our responses have adequately addressed all the comments. Thank you again for your consideration.

**Response to Reviewer #1's comments**

The revised manuscript is much improved. Well done by the authors. While it could be on track for publication, there are a few important considerations and perhaps one methodological error that must first be addressed. Please see my comments below, which are intended to be positive and constructive. I would be happy to review a revised version of the manuscript and I again commend the authors for the great progress they have made with this manuscript.

We greatly appreciate the reviewer's positive assessment of our (first) revision and manuscript. We have made substantial changes to our manuscript and hope that our responses have adequately addressed all the comments.

**Major comments:**

1. What were the units for discharge when you modeled constituent fluxes with LOADEST? From your SI dataset, it seems that you used $m^3$ $s^{-1}$? As detailed in Runkel et al. (2004),

LOADEST takes discharge in units of $ft^3\ s^{-1}$. If you used $m^3\ s^{-1}$, then your constituent modeling must be updated and associated values throughout the text corrected (e.g., Sec. 3.1). Of course, this would be relatively easy to do! This being said, I agree with reporting discharge in metric units ($m^3\ s^{-1}$) in the main text (e.g., L210), as is commonly done.

Runkel, R. L., Crawford, C. G., & Cohn, T. A. (2004). *Load Estimator (LOADEST): A FORTRAN program for estimating constituent loads in streams and rivers* (No. 4-A5).
https://pubs.usgs.gov/tm/2005/tm4A5/pdf/508final.pdf

Thank you! Our original phrase may be vague. The units for discharge were actually cubic feet per second ($ft^3\ s^{-1}$) when we modelled constituent fluxes with LOADEST. The daily discharge is reported in metric units ($m^3\ s^{-1}$) in the main text and SI dataset. The above explanations are added in *Line 196*.

2. Following on Dr. Park's comment about headwater streams: Headwaters are generally the smallest surficial fluvial component within a stream network. I read Meyer and Wallace (2001) and could not find where they explicitly state that headwaters include $1^{st}$ to $3^{rd}$ order streams, as your citation in L34 implies… but please feel free to prove me wrong about this! The lowest stream order is not necessarily a headwater stream, because definitions of stream order are relative and can vary depending on the resolution of the data used to determine stream order. For instance, stream order derived from geospatial data like a DEM (often relatively coarse, e.g., 90 m from Lin et al., 2019 WRR) may be quite different than stream order determined by an expert in the field or using high-resolution satellite imagery to visually map streams. From Google Earth imagery and your Figure 1, sampling points SLH2–5 do not appear to be headwater streams, and I am not convinced that SLH0–1 are, either. All this to say, I am not convinced that the streams in your study are truly headwater streams and I would not use GIS-derived estimates of stream order (or the citation to Meyer and Wallace 2001) to try to assert this. I think it would be appropriate to place less emphasis on interpreting your results as direct measurements of headwater streams. I think this does not require a major rewrite of your paper and can be addressed without too much trouble by modifying the very nice text that you have in the Introduction and elsewhere. For instance, you could reason that your sampling points capture the integrated hydrochemical signal across headwater catchments. This would be compelling and more accurate. Clarifying this, and omitting 'headwater' where appropriate, would more accurately represent your study sites without comprising the overall relevance of your research, or its significance for understanding hydrochemical signals within headwaters of the QTP.

Good point! Definition of headwater streams is a long-standing conundrum for researchers. Admittedly, we made a mistake in citation of the correlation of headwater stream with stream order. Headwater streams are considered to be first- and second-order streams (Meyer and Wallance, 2001), while they are grouped into first- to third-order streams in the report of Vannote et al. (1980). Anyway, use of stream order to define headwater streams is problematic because, as you mentioned here, stream-order designations vary depending upon the accuracy and resolution of the stream delineation.

Given the lack of a universal and spatially-explicit definition of headwater streams, we revise the texts on the definition of headwater streams, the description of our study site, and the discussion of Shaliu River as "a typical headwater stream". In addition, the title is revised to "spatial-temporal variations in riverine carbon strongly influenced by local hydrological events in an alpine headwater catchment", because our study site captures the integrated signal across headwater catchments (rather than a stream within a narrow channel). Detailed revisions are in *Lines 17, 28, 31, 34, 71, 84, 88, 91-92, 396, 405*. We sincerely appreciate your considerate comments!

Reference:

Meyer, J. L. and Wallace, J. B.: Lost linkages and lotic ecology: rediscovering small streams, Ecology: Achievement and Challenge, The 41[st] Symposium of the British Ecological Society of America, 295-317 pp.2001.

Vannote, R. L., Minshall, G. W., Cummins, K. W., Sedell, J. R., and Cushing, C. E.: River continuum concept, Can. J. Fish. Aquat. Sci., 37, 130-137, https://doi.org/10.1139/f80-017, 1980.

**Minor comments:**

1. Also related to LOADEST: I see now (from the text in Sec. 2.5 and Table S1) that you consider all possible LOADEST regression equations when modeling C flux. i.e., It appears that you selected Model 0 and let LOADEST automatically choose the most parsimonious model (lowest AIC) from models 1–9. However, model #s 3, 5, 7–9 include covariates for long-term change, which are arguably more appropriate to use when you have measurements from across multiple years (even five years of observations is not considered long enough by some studies, e.g., Zolkos et al., 2020). Because your study period is not multi-annual, and you are not investigating long-term trends, it would be more appropriate to estimate fluxes using the best model from 1, 2, 4, or 6.

Zolkos, S., Krabbenhoft, D. P., Suslova, A., Tank, S. E., McClelland, J. W., Spencer, R. G., ... & Holmes, R. M. (2020). Mercury Export from Arctic Great Rivers. *Environmental Science & Technology*, *54*(7), 4140-4148.

Thank you! The more appropriate LOADEST regression model with lowest Akaike Information Criterion (AIC) and without variables for long-term change during the calibration period (i.e., Model 1, 2, 4, 6; Zolkos et al., 2020) is re-selected to estimate fluxes. Relevant revision is added in *Lines 190-192, Line 195, Figure 2 and Table S1*.

**Additional comments:**

L14: "dynamics" is vague. Avoiding it generally improves clarity. If there is more to the story than just C 'transport' / fluxes, consider clarifying how 'dynamics' relates to it.

The word "dynamics" is revised to "fluxes", "transport", or "spatial-temporal variations" as needed. Please refer to the track change version of manuscript for details.

L17: What do you mean by "in-depth"? Please clarify.

The word "in-depth" means that we conducted sampling at a high spatial resolution in both pre-monsoon and monsoon seasons. This is clarified in *Line 18*.

L22: "… thawed frozen…". Are the soils thawed or frozen? I see what you are trying to say here. It would be clearer to just say 'thawed'.

Revised in *Line 22*. Thank you!

L30: "dynamics"- see comment for L14.

Revised in *Line 30*.

L30: What exactly do you mean by "cascading effects"? i.e., that thaw effects in headwaters might propagate downstream, across watershed scales? (especially for dissolved inorganic carbon, in your region) If so, you might consider recent work from the western Canadian Arctic exploring this topic: https://bg.copernicus.org/articles/17/5163/2020/bg-17-5163-2020.html, but also recognize that thaw effects on fluvial C cycling (especially DIC) across watershed scales vary across permafrost regions. For Siberia, see: https://agupubs.onlinelibrary.wiley.com/doi/full/10.1029/2017JG004311. Perhaps worth considering in the Discussion, and how your work adds to this understanding.

Our original phrase might be misleading. We actually mean that carbon variations in headwater

streams might critically affect the biogeochemical cycles of the downstream watersheds (such as high-order rivers) when facing permafrost thawing and precipitation events. This phrase is revised at *Lines 30-31* and *Line 404* as follows:

*"...thawing of permafrost and alterations of precipitation regimes may significantly influence the alpine headwater carbon transport, with critical effects on the biogeochemical cycles of the downstream rivers."*

L63-4: Why is it important that there are different C sources? It would be interesting and helpful to clarify the biogeochemical relevance of the C sources, and doing so could help to justify the rationale for your Ad/Al ratio measurements. For instance, do the sources have varying degrees of recalcitrance?

Good point! It is reported that the aged permafrost carbon and modern carbon released following permafrost thawing may have different degradation potentials (Mann et al., 2015), and this sentence is added in *Lines 63-64*.

Reference:

Mann, P. J., Eglinton, T. I., McIntyre, C. P., Zimov, N., Davydova, A., Vonk, J. E., Holmes, R. M., and Spencer, R. G. M.: Utilization of ancient permafrost carbon in headwaters of Arctic fluvial networks, Nat. Commun., 6, 7, https://doi.org/10.1038/ncomms8856, 2015.

L89: The value for mean annual daily discharge is interesting. It would also be interesting to know total annual discharge (e.g., $km^3 \, y^{-1}$), from your daily Q measurements.

According to our daily discharge measurements, a total annual discharge of $2.4 \times 10^8 \, m^3$ during 2015 to 2016 is estimated and added in *Line 213*.

L90-1: Please see my Major comment #2.

Revisions on headwater stream is shown in *Lines 17, 28, 31, 34, 71, 84, 88, 91-92, 396, 405*.

L95-6: No carbonate lithologies? Why are your DIC concentrations so high? This should be discussed a bit more. My recollection is that some of the recent work by Song et al. discusses carbonate lithologies on the QTP.

Thank you! Our prior description on bedrocks in the Shaliu River is incomplete because Zhang et al. (2013) mainly focused on the lithology comparison between Buha and Shaliu River and their effects on silicate weathering. However, besides the Triassic sandstone, late Cambrian metamorphic rocks (schist and gneiss) and granites, there are actually abundant late Paleozoic marine sedimentary rocks rich in carbonates (Bissell and Chilingar, 1967) in the upper and lower reaches of the Shaliu River when searching more related literatures (Jin et al., 2009; Xiao et al., 2012; Xiao et al., 2013). To keep consistency with our subject on carbon transport, we revise the lithology characteristics by adding "late Paleozoic marine sedimentary rocks" in *Lines 97-98*.

Reference:

Zhang, F., Jin, Z. D., Li, F. C., Yu, J. M., and Xiao, J.: Controls on seasonal variations of silicate weathering and $CO_2$ consumption in, two river catchments on the NE Tibetan Plateau, J. Asian Earth Sci., 62, 547-560, https://doi.org/10.1016/j.jseaes.2012.11.004, 2013.

Bissell, H.J., Chilingar, G.V.: Classification of sedimentary carbonate rocks. In: Developments in Sedimentology, vol. 9, pp. 87–168, 1967.

Jin, Z. D., Yu, J. M., Wang, S. M., Zhang, F., Shi, Y. W., and You, C. F.: Constraints on water chemistry by chemical weathering in the Lake Qinghai catchment, northeastern Tibetan Plateau (China): clues from Sr and its isotopic geochemistry, Hydrogeol. J., 17, 2037-2048, https://doi.org/10.1007/s10040-009-0480-9, 2009.

Xiao, J., Jin, Z. D., and Zhang, F.: Geochemical and isotopic characteristics of shallow groundwater within the Lake Qinghai catchment, NE Tibetan Plateau, Quatern. Int., 313, 62-73, https://doi.org/10.1016/j.quaint.2013.05.033, 2013.

Xiao, J., Jin, Z. D., Zhang, F., and Wang, J.: Major ion geochemistry of shallow groundwater in the Qinghai Lake catchment, NE Qinghai-Tibet Plateau, Environ. Earth Sci., 67, 1331-1344, https://doi.org/10.1007/s12665-012-1576-4, 2012.

L202: Please specify: What post-hoc test did you use?

*The "post-hoc Duncan test" is added in Line 205.*

L239: Would be good to consistently include valencies for calcium and magnesium throughout, as you do here (e.g., L156).

*Revised in Lines 157-158.*

L307: Pardon the editorial comment, but it would be more appropriate for dissolved 'bicarbonates' and 'carbonates' to be singular, not plural.

*Revised in Line 307.*

L311-12: Please see my comment for L95-6. Are there carbonate lithologies? Consistency in the message would help.

*Carbonate lithologies are added in Lines 311-312. Thank you!*

L313: Technically, high $Ca^{2+}$ and $Mg^{2+}$ concentrations and a strong association with DIC reflect greater availability of carbonate-bearing lithologies for chemical weathering, not necessarily high rates of weathering. So, I am not sure that I would say "proved" here. Perhaps say "reflected" or "indicated by".

*The word "proved" is revised to "reflected" in Line 314.*

L318-320: Rainfall should enhance carbonate weathering, no? (e.g., Zeng et al., 2019) Perhaps clarify and elaborate on this.

Zeng, S., Liu, Z., & Kaufmann, G. (2019). Sensitivity of the global carbonate weathering carbon-sink flux to climate and land-use changes. *Nature Communications*, *10*(1), 1-10.

*Good point! We do agree that increasing rainfall should enhance carbonate weathering carbon-sink flux (CCSF) as reported by Zeng et al. (2019), since fluxes are calculated as the function of alkalinity concentration and discharge. This discharge (or rainfall) effects on DIC flux is displayed in our study as well (Figure 2b). However, it is more complicated when referring to rainfall effects on DIC (alkalinity or bicarbonate) concentration. Previous studies show that increased discharge (due to increasing rainfall) could enhance alkalinity concentration yet at five-decade scale (Raymond and Cole, 2003) or at north-south profile through humid belts (Strakhov, 1967). By contrast, instantaneous alkalinity concentration is negatively correlated with discharge (Raymond and Cole, 2003 and Figure 3 therein) or precipitation (Figure R1; Zeng et al., 2016) at short-term or small-region scales, which is normally attributed to a pronounced dilution effects of rainfall (Raymond and Cole, 2003; Zeng et al., 2016) due to the lower DIC concentration in rainwater than stream water (Song et al., 2019). Since our study is conducted at a relatively short-term scale during 2015 to 2016, we postulate that the negative relationship between DIC concentration and discharge is caused by a dilution effect. The above explanation is added in Lines 320-321.*

[Figure]

**Figure R1.** Relationship between mean annual precipitation and bicarbonate equilibrium concentration in karst region (seven provinces) of Southwestern China from 1970s to 2010s. Modified from Zeng et al. (2016).

Reference:

Zeng, S. B., Liu, Z. H., and Kaufmann, G.: Sensitivity of the global carbonate weathering carbon-sink flux to climate and land-use changes, Nat. Commun., 10, 10, https://doi.org/10.1038/s41467-019-13772-4, 2019.

Zeng, S. B., Jiang, Y. J., and Liu, Z. H.: Assessment of climate impacts on the karst-related carbon sink in SW China using MPD and GIS, Global Planet. Change, 144, 171-181, https://doi.org/10.1016/j.gloplacha.2016.07.015, 2016.

Raymond, P. A. and Cole, J. J.: Increase in the export of alkalinity from North America's largest river, Science, 301, 88-91, https://doi.org/10.1126/science.1083788, 2003.

Song, C. L., Wang, G., Mao, T. X., Chen, X. P., Huang, K. W., Sun, X. Y., and Hu, Z. Y.: Importance of active layer freeze-thaw cycles on the riverine dissolved carbon export on the Qinghai-Tibet Plateau permafrost region, PeerJ, 7, 25, https://doi.org/10.7717/peerj.7146, 2019.

Strakhov, N.M.: Principles of lithogenesis, vol.1, Tomkeieff S.I., Hemingway J.E., Eds, Oliver & Boyd, London, 1967.

L352: "headstream"- i.e., "headwater stream"?

The word "headstream" is revised to "stream" in *Line 353*.

L398: You could simply and clarify by saying "DIC" instead of "dissolved carbon (especially DIC)".

Revised in *Line 397*.

L401: Suggest deleting "frozen" (related to my comment for L22).

Deleted in *Line 399*.

L402: "levels" = "concentrations"?

Revised in *Line 400*.

L402: Apparently higher sensitivity of DIC concentration than DOC concentration, but also very interesting that DOC source appears to change. Worth mentioning here.

This sentence is revised accordingly in *Lines 401-402*.

Figure 2: Are the y-axis units metric 'tonnes' per day, rather than imperial 'tons'? For easier

comparison with other figures and values, it would help state the y-axis in metric units (e.g. Megagrams, Mg, or in grams if desired) and re-scale as needed.

The y-axis units in Figure 2b are revised to Megagrams carbon per day (Mg C day$^{-1}$).

---

## Author Response (AR3)

**Response to Reviewers of bg-2020-472**

**General response to the Editor**

**Associate Editor's comments:**

Dear Authors,

I thank you for incorporating the very constructive reviewer comments in your second revision. I am pleased to let you know that your manuscript can be published in Biogeosciences, after some technical corrections as follows:

[*Detailed comments were shown below…*]

It has been a great pleasure for me to take part in the review process where you and the reviewers have done excellent jobs in improving the presentation and interpretation of invaluable data obtained from a hard-to-access alpine river system. In particular, I would like to express my sincere thanks to the anonymous reviewer who offered two detailed reports containing very constructive comments, which is rare in this age of "quick reviews", and thereby warranting authors' acknowledgement.

Sincerely,

Ji-Hyung Park

Associate Editor, Biogeosciences

We sincerely thank the editor and reviewers for the considerate and constructive comments on improving this manuscript during the entire review process. We have made substantial corrections to our manuscript this time (details below), and hope our responses have adequately addressed all the comments.

**Detailed comments:**

- Consistent use of headwater streams and catchments: While you tried to limit the use of headwater streams to small-order streams that could not be identified easily on the coarse-resolution map in Fig. 1, there are still some confusing statements including:

- 1. Title: "an alpine headwater catchment" is Shaliu River, so the phase needs to be rewritten (like "an alpine catchment") to avoid any confusion.

Revised.

- 2. Abstract: "an alpine small stream (the Shaliu River capturing headwaters within its catchment)" can be changed to something like "the Shaliu River, an alpine small river receiving many headwater streams"

Revised.

- 3. Main text (lines 71, 84, 91-92,,,): Please use the terms "river" and "headwater streams" consistently. For instance, the lines 91-92 can be rewritten (The Shaliu River is a small river integrating headwater streams of orders of $1-3$). To reflect these changes, you may need to change the Fig. 1 legend that still allocates 1-3 orders to the entire reaches. Please refer again to the reviewer comment.

The terms "river" and "headwater streams" are revised in the main text according to the context, and Figure 1 is revised as follows:

[Figure]

*Figure 1. Sampling sites along the Shaliu River (a) and soil temperature (T) at the depth of 10 cm (referred to as topsoil) and 40 cm (subsoil) at SLH-1 and SLH-3 stations (b). The map in panel (a) is processed with ArcGIS 10.0. The blue and red arrows in panel (b) indicate the sampling time for soil solution at SLH-1 and SLH-3 during thawing event, respectively.*